# Scalable Bayesian Inference for Nonlinear Conservation Laws

**Tim Weiland** [1]   **Philipp Hennig** [1]

## Abstract

Nonlinear conservation laws are at the heart of many of the most important dynamical systems in science and engineering. In practical applications, such systems are often subject to various sources of uncertainty, e.g. due to sparse or noisy measurements. Inferring physical quantities and fields of interest then becomes an ill-posed problem which both classical numerical methods and modern deep learning-based methods struggle to treat appropriately. Recent work has framed classical numerical methods as Bayesian inference under Gaussian process priors, resulting in a physics-aware treatment of uncertainties. Following this line of work, we develop a novel numerically conservative method for uncertainty-aware simulations of nonlinear conservation laws. We use recent sparse approximation techniques to scale up to large-scale forward and inverse problems. For forward simulation, we inherit the accuracy of classical solvers while providing structured uncertainty quantification. On inverse problems, we recover posteriors over nonparametric source fields in seconds — outperforming neural baselines that take minutes to produce a less accurate point estimate.

## 1. Introduction

Conservation laws are a subclass of partial differential equations (PDEs) with applications ranging from weather prediction (Lauritzen et al., 2011) and aerodynamics (Vos et al., 2002) to flood modeling (Kurganov, 2018) and subsurface contaminant transport (Bear, 1988). They take the general form

$$\frac{\partial u}{\partial t} + \nabla \cdot F(u) = s, \qquad (1)$$

where $u$ denotes the conserved quantity, $F(u)$ the (in general nonlinear) flux operator, and $s$ the source term.

**Real-world applications incur uncertainty.** In the classical *forward problem*, $u$ is the object of interest: Under initial and boundary conditions, it is inferred from the known flux and source terms. Practical applications however often operate in settings that lack information: Sensors measuring relevant quantities are noisy and often only available at sparse locations for cost reasons. Worse, in many applications the source itself is unknown—a contamination event must be localized, or an emission field reconstructed, from sparse measurements of $u$. These *inverse problems* admit multiple solutions consistent with the data. Orthogonal to these issues, any solver used to approach these problems may only produce an approximate solution. Together, these factors demand methods that quantify uncertainty – whether epistemic, computational, or otherwise – to help practitioners make informed decisions.

**In the machine learning community**, neural approaches such as physics-informed neural networks (Raissi et al., 2019) and neural operators (Kovachki et al., 2023) have gained popularity for PDE solving. While powerful, they are point estimators by design. Ensembles or Bayesian variants may quantify uncertainty, but at significant computational cost and often with poor calibration (Podina et al., 2024).

**Probabilistic numerical methods** frame numerical methods as Bayesian inference, thereby treating uncertainty naturally while matching the accuracy of classical methods in the mean (Hennig et al., 2022). Recent work has developed such methods for the solution of PDEs by imposing a Gaussian process (GP) prior over the object(s) of interest and conditioning on physical information (Cockayne, 2019; Pförtner et al., 2024). While this is principled, GP inference naively scales cubically in the discretization resolution. These approaches are mostly based on *collocation*, which corresponds to a pointwise discretization of the PDE.

**Numerical conservation.** Conservation laws derive their name from a defining property: quantities such as mass or energy are neither created nor destroyed. Numerical methods that fail to preserve this can produce unphysical solutions that drift over time. The *finite volume method* (FVM) (Eymard et al., 2000) enforces conservation by construction, discretizing the integral balance rather than the

[1]Tübingen AI Center, University of Tübingen, Tübingen, Germany. Correspondence to: Tim Weiland <tim.weiland@uni-tuebingen.de>.

*Proceedings of the 43${}^{rd}$ International Conference on Machine Learning*, Seoul, South Korea. PMLR 306, 2026. Copyright 2026 by the author(s).

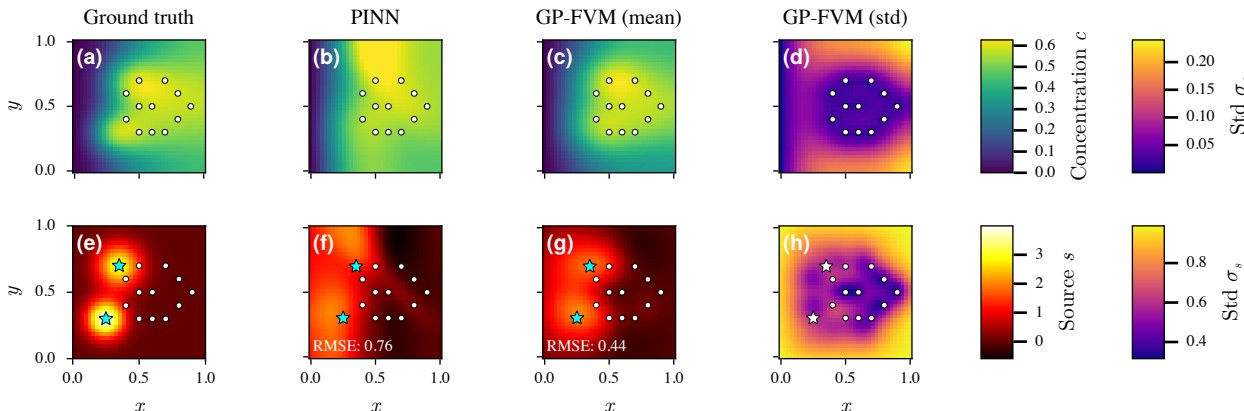

*Figure 1.* **Source identification from sparse sensor measurements.** Top row: concentration field. Bottom row: inferred source field. Our GP-FVM method (right) recovers the source locations with quantified uncertainty, while a PINN baseline produces a point estimate with artifacts. Our method solves this nonparametric problem in seconds; the PINN requires minutes.

PDE pointwise — making it the standard in computational fluid dynamics. Expressing FVM as a scalable probabilistic numerical scheme has remained an open problem.

**Recent work** has begun addressing scalability for GP-based PDE solvers. Chen et al. (2025) apply a Vecchia approximation to solve nonlinear PDEs, but via collocation. Weiland et al. (2024) develop a probabilistic FVM, but their method is restricted to linear PDEs and their low-rank approximation does not yield reliable UQ at large scales.

**Our contributions.** We present a scalable method for Bayesian inference on nonlinear conservation laws. We formulate the finite volume method as inference under a Gaussian process prior — extending prior work on linear PDEs to nonlinear flux operators through iterative conditioning. To achieve scalability, we extend Vecchia approximations to integral observations and introduce a sparsity-preserving time-stepping scheme. Our framework can solve forward and inverse problems alike, while propagating uncertainties and explicitly embedding prior knowledge.

On a source identification inverse problem (Figure 1), our method recovers nonparametric source fields with quantified uncertainty in seconds, outperforming a physics-informed neural network baseline that requires minutes for a less accurate point estimate. Section 2 develops the high-level method, Section 3 addresses scalability, and Section 4 presents experiments.

## 2. Probabilistic Conservation Schemes

Our key insight is that the gold standard discretization for conservation laws, namely the finite volume method (FVM), may be expressed as Bayesian inference under a GP prior. We first review how GPs interact with linear functionals (Section 2.1), then recall the classical FVM discretization

(Section 2.2), and finally show how to unify them into a coherent Bayesian framework (Section 2.3).

### 2.1. Gaussian Processes and Linear Functionals

We model our prior knowledge about the solution function $u$ in Equation (1) through a Gaussian process (GP) prior with mean function $\mu$ and kernel function $k$:

$$u \sim \mathcal{GP}(\mu, k). \tag{2}$$

In the following, we discuss how to express the simulation of conservation laws as Bayesian inference under this prior.

We leverage that linear transformations of Gaussian processes remain Gaussian: If $\ell$ is a bounded linear functional acting on $u$, then $\ell(u)$ is a Gaussian random variable with

$$\ell(u) \sim \mathcal{N}\big(\ell\mu,\, \ell k\ell'\big), \tag{3}$$

where $\ell\mu := \ell[\mu(\cdot)]$ denotes the functional applied to the mean, and $\ell_1 k\ell_2' := \ell_1[x \mapsto \ell_2[k(x, \cdot)]]$ denotes the functionals applied to each argument of the kernel. More generally, given a collection of linear functionals $\ell_1, \ldots, \ell_n$, the vector $[\ell_1(u), \ldots, \ell_n(u)]^\top$ is jointly Gaussian:

$$\begin{bmatrix} \ell_1(u) \\ \vdots \\ \ell_n(u) \end{bmatrix} \sim \mathcal{N}\left( \begin{bmatrix} \ell_1\mu \\ \vdots \\ \ell_n\mu \end{bmatrix}, \begin{bmatrix} \ell_1 k\ell_1' & \cdots & \ell_1 k\ell_n' \\ \vdots & \ddots & \vdots \\ \ell_n k\ell_1' & \cdots & \ell_n k\ell_n' \end{bmatrix} \right). \tag{4}$$

As a concrete example, consider a GP prior over functions $u : [0, 1] \to \mathbb{R}$. We can define an evaluation functional $\ell_1[u] := u(0.5)$ that extracts the function value at the midpoint, and a second-derivative functional $\ell_2[u] := u''(0.5)$ that extracts the curvature. Applying Equation (4), the pair $\boldsymbol{s}_{\text{example}} := (u(0.5), u''(0.5))^\top$ is jointly Gaussian.

Since Gaussians are closed under linear conditioning, we can incorporate observations in closed form. Let $\boldsymbol{s} \in \mathbb{R}^N$

be Gaussian with mean $\boldsymbol{\mu} \in \mathbb{R}^N$ and precision matrix $\boldsymbol{Q} \in \mathbb{R}^{N \times N}$, i.e. $\boldsymbol{s} \sim \mathcal{N}(\boldsymbol{\mu}, \boldsymbol{Q}^{-1})$. Here $N$ denotes the dimension of the joint state $\boldsymbol{s}$. Given a matrix $\boldsymbol{A} \in \mathbb{R}^{M \times N}$ and noisy observations $\boldsymbol{As} \sim \mathcal{N}(\boldsymbol{b}, \sigma^2 \boldsymbol{I})$, define $\hat{\boldsymbol{Q}} := \boldsymbol{Q} + \sigma^{-2} \boldsymbol{A}^\top \boldsymbol{A}$. The posterior $\boldsymbol{s}^* := (\boldsymbol{s} \mid \boldsymbol{As} = \boldsymbol{b})$ is

$$\boldsymbol{s}^* \sim \mathcal{N}\big(\hat{\boldsymbol{Q}}^{-1}(\boldsymbol{Q}\boldsymbol{\mu} + \sigma^{-2} \boldsymbol{A}^\top \boldsymbol{b}), \ \hat{\boldsymbol{Q}}^{-1}\big). \quad (5)$$

Returning to our example: We can e.g. set $\boldsymbol{A} = \begin{bmatrix} 0 & 1 \end{bmatrix}$ (the row vector selecting $u''(0.5)$ from $\boldsymbol{s}_{\text{example}}$) and $\boldsymbol{b} = 0$ in Equation (5) to obtain a posterior of $\boldsymbol{s}_{\text{example}}$ that reflects $u''(0.5) \approx 0$. We could extend our state by many more, spread-out points at which we evaluate the second derivative (so-called *collocation points*), which would allow us to approximate a solution to the 1D Laplace equation

$$u''(x) = 0. \quad (x \in (0, 1)) \quad (6)$$

Physics-informed GP regression (Pförtner et al., 2024) develops this idea systematically: boundary conditions, PDE discretizations and sensor measurements are all just sources of information (in the sense of Cockayne (2019)), and conditioning yields a posterior that unifies them.

## 2.2. Finite Volume Discretization

The Finite Volume Method (FVM) discretizes Equation (1) by integrating over control volumes $\Omega_1, \ldots, \Omega_N$:

$$\frac{\partial}{\partial t} \int_{\Omega_i} u \, dV + \int_{\Omega_i} \nabla \cdot F(u) \, dV = \int_{\Omega_i} s \, dV. \quad (7)$$

The divergence theorem yields:

$$\frac{\partial}{\partial t} \int_{\Omega_i} u \, dV + \oint_{\partial\Omega_i} F(u) \cdot \mathbf{n} \, dS = \int_{\Omega_i} s \, dV. \quad (8)$$

**Classical numerical FVM** schemes represent the solution in terms of one or multiple nodes per cell, and then approximate these integrals numerically. In the simplest setting, a piecewise constant representation is used. The compute nodes of FVM then are the cell midpoints $\bar{u}_i$, and the integral over each cell is approximated by the midpoint rule

$$\int_{\Omega_i} u \, dV \approx |\Omega_i| \bar{u}_i. \quad (9)$$

The integral of the source function $s$ as well as the surface integral of the flux $F$ are similarly approximated through numerical quadrature. The latter requires a scheme to extrapolate or interpolate $F$ at the cell surface from the compute nodes, for which many different *flux schemes* exist.

From the discretization of Equation (1) to the use of quadrature and flux schemes, FVM makes various approximations. Next, we will derive a GP-based FVM approach that naturally expresses these approximation errors as uncertainty.

## 2.3. FVM as Bayesian Inference

In the following, we express the components in Equation (8) in terms of linear functionals. We then obtain a joint Gaussian prior through the push-forward of GP priors under these linear functionals. Then fulfilling Equation (8) reduces to (non-)linear conditioning of this prior.

**Integration is linear.** Recall that integration is a linear operation, and thus $\ell_{vol_i}[u] := \int_{\Omega_i} u \, dV$ is a linear functional. One might similarly define $\ell_{\text{flux}_i}[u] := \oint_{\partial\Omega_i} F(u) \cdot \mathbf{n} \, dS$, as done by Weiland et al. (2024). But this is only a *linear* functional if $F$ is linear, which is not the case for nonlinear PDEs. Thus, the flux term warrants a different approach.

**Flux term.** Consider as example the flux for the inviscid Burgers' equation: $F_{\text{Burgers}}(u) = \frac{1}{2}u^2$. In *one* spatial dimension, the cells are intervals $\Omega_i = [b_i, b_{i+1}]$, and so

$$\oint_{\partial\Omega_i} F(u) \cdot \mathbf{n} \, dS = F(b_{i+1}) - F(b_i).$$

Thus, we only need linear functionals for the components that make up $F$ at the cell boundaries. Concretely, for $F_{\text{Burgers}}$, we may use $\ell_{\text{boundary}_i}[u] = u(b_i)$.

In 2D and 3D, we need to compute actual line and surface integrals of the flux. We propose a two-stage approach to this. First, we model the flux field with its own prior $F(u) \sim \mathcal{GP}(\mu_F, k_F)$, treated initially as a quantity independent of $u$. The deterministic relationship $F = F(u)$ imposed by the PDE is then enforced through nonlinear conditioning of pointwise evaluations of $F$ on pointwise evaluations of $u$ (detailed below), yielding a posterior that respects the actual flux–solution coupling. Afterwards, the flux term is directly a linear functional in $F(u)$: $\tilde{\ell}_{\text{flux}_i}[F(u)] := \oint_{\partial\Omega_i} F(u) \cdot \mathbf{n} \, dS$. This amounts to per-cell Bayesian quadrature (O'Hagan, 1991).

**Source term.** Some commonly presented conservation laws have constant (often zero) source functions, in which case the treatment of the source term in Equation (8) is trivial. If the source function is non-constant (or even unknown), we may employ the same method as for the flux term: Model the source with a GP prior $s \sim \mathcal{GP}(\mu_s, k_s)$, incorporate point observations of the source (if available), and finally define linear functionals $\ell_{\text{source}_i}[s] := \int_{\Omega_i} s \, dV$.

**Additional linear functionals** are used to embed information on top of the FVM constraints:

- Initial and boundary conditions provide point observations of the solution $u$.
- Point observations of the flux may be indirectly obtained through observations of $u$.
- Point observations of the source are either available or the object of interest (in an inverse problem setting).

We add these linear functionals to the joint state. For no-

tational simplicity, we write $\boldsymbol{\ell}_u : U \to \mathbb{R}^{N+M_u}$ for the vector-valued functional stacking the $N$ FVM cell-integral functionals $\ell_{vol_i}$ together with the $M_u$ functionals encoding additional information about $u$; the stacked flux functional $\boldsymbol{\ell}_F$ (collecting the per-cell $\tilde{\ell}_{\text{flux}_i}$) and stacked source functional $\boldsymbol{\ell}_s$ (collecting $\ell_{\text{source}_i}$) are defined analogously.

**Putting it all together.** We have independent GP priors for the solution, flux, and source. This results in the joint state

$$\boldsymbol{s} := \begin{bmatrix} \boldsymbol{\ell}_u[u] \\ \boldsymbol{\ell}_F[F(u)] \\ \boldsymbol{\ell}_s[s] \end{bmatrix} \sim \mathcal{N}\left( \begin{bmatrix} \boldsymbol{\mu}_u \\ \boldsymbol{\mu}_F \\ \boldsymbol{\mu}_s \end{bmatrix}, \begin{bmatrix} \boldsymbol{\Sigma}_u & & \\ & \boldsymbol{\Sigma}_F & \\ & & \boldsymbol{\Sigma}_s \end{bmatrix} \right), \tag{10}$$

where each block is given by Equation (4).

Now, we can express our various sources of information in terms of conditioning. Initial and boundary conditions, source observations, and even the FVM constraints in Equation (8) are all linear combinations of state components. We obtain a posterior conditioned on these observations through one application of Equation (5). Though their priors are independent, this conditioning couples all three quantities in the posterior.

The only possible source of nonlinearity is the flux $F(u)$: although $F$ has its own GP prior, the flux–solution map $F = F(u)$ itself is a deterministic nonlinear relationship that still needs to be enforced. Concretely, the joint state contains pointwise evaluations of both $F(u)$ and $u$ at points $\boldsymbol{x} := (x_1, \ldots, x_k)^\top$. We want to enforce a nonlinear relationship $f(u(\boldsymbol{x}), F(u)(\boldsymbol{x})) = \boldsymbol{0}$. To do so, we find the maximum a posteriori (MAP) estimate of the joint state $\boldsymbol{s}$ (with the constraint locations $\boldsymbol{x}$ fixed) under the likelihood

$$\boldsymbol{y} \mid \boldsymbol{s} \sim \mathcal{N}(f(u(\boldsymbol{x}), F(u)(\boldsymbol{x})), \sigma^2 \boldsymbol{I}), \tag{11}$$

with observations $\boldsymbol{y} = 0$. Gaussian likelihoods on zero-valued observations of this form have appeared in related probabilistic modeling contexts as *virtual likelihoods* or *virtual observables* for softly enforcing physical constraints (Kaltenbach & Koutsourelakis, 2020). We find the MAP estimate via Gauss-Newton optimization and then use a Laplace approximation to obtain a Gaussian posterior.

## 2.4. Implications

**Unified inference.** Our method forms a joint prior over solution, flux and source. This enables us to unify different sources of information (i.e. conservation law, boundary conditions, sensor measurements) in one common language. In addition, we clearly express our structural assumptions about each term through the prior. The result is a framework that handles both forward and inverse problems natively by simply changing what is observed vs. inferred:

| Problem | Observed | Inferred |
|---|---|---|
| Forward | $s, F$ | $u$ |
| Source identification | $u, F$ | $s$ |
| Flux identification | $u, s$ | $F$ |

The FVM constraints create coupling between the three quantities that propagate the different information sources. This concept of maintaining a joint distribution over the solution trajectory and latent force(s) has been explored in the context of ODE solvers by Schmidt et al. (2021).

**Differences to classical FVM.** Where classical FVM uses numerical quadrature, we implicitly employ Bayesian quadrature and thus propagate quadrature error to the joint uncertainty. For classical FVM, there are schemes to achieve higher convergence orders for suitable problems. In our framework, as will be demonstrated in Section 4.3, this may be tweaked effortlessly through the smoothness of the kernel. For classical FVM, there is a whole zoo of flux schemes to arrive at the flux values from the solution function values. In our method, there is no *separate* flux scheme: The flux term is automatically derived from conditioning the flux prior on observations of the solution.

**Computational bottleneck.** The key challenge of our method is that it quickly produces a large total state dimension $N$, and Gaussian inference naively scales as $\mathcal{O}(N^3)$ compute. Hence, Section 3 will explore in detail techniques to implement our conceptual framework efficiently.

## 3. Scalable Inference

For a scalable implementation of the concepts explored in Section 2, we propose an approach based on a structured prior (Section 3.1), sparse inference algorithms (Sections 3.2 and 3.3) and a careful treatment of time (Section 3.4).

### 3.1. A Structured Prior

We employ a separable prior with tensor product structure. For the spatial dimensions, we use a tensor product of Matérn kernels:

$$k_{\text{space}}(\boldsymbol{x}, \boldsymbol{x}') = \prod_{i=1}^d k_{\text{Mat}}^{(\nu_i, \lambda_i)}(x_i, x_i'), \tag{12}$$

where $k_{\text{Mat}}^{(\nu, \lambda)}$ denotes the Matérn kernel with smoothness $\nu$ and lengthscale $\lambda$. This structure enables closed-form evaluation of integrals (Briol et al., 2025) and partial derivatives needed for Equation (4), and allows independent control of smoothness in each coordinate. To obtain closed-form kernel integrals, we use rectangular FVM cells. We deliberately use Matérn rather than e.g. squared-exponential (SE) kernels: the sparse Cholesky approximation in Section 3.3 relies on the screening effect, which holds for finitely smooth

kernels but is known to degrade for the infinite smoothness of the SE kernel (Schäfer et al., 2021).

For the temporal dimension, we use an Integrated Wiener Process (IWP) of order $q$ (Schober et al., 2019). The IWP encodes that the $q$-th time derivative is Brownian motion, naturally capturing temporal evolution. A side effect is that the time derivative $\frac{\partial}{\partial t}$ needed for Equation (8) is built into the state representation. Importantly, the IWP is non-stationary, so conservation properties are determined entirely by the FVM constraints. A stationary prior (e.g., temporal Matérn) reverts toward its mean, which cancels the numerical conservation of energy of FVM.

With this, the full spacetime kernel is

$$k\big((t, \boldsymbol{x}), (t', \boldsymbol{x}')\big) = k_{\text{IWP}}^{(q)}(t, t') \cdot k_{\text{space}}(\boldsymbol{x}, \boldsymbol{x}'). \quad (13)$$

Note that separability is a property of the *prior*, not of the posterior. The FVM constraints in Equation (8) couple space and time through conditioning, so the posterior is generally non-separable — reflecting the actual spacetime structure imposed by the PDE. A similar assumption is made in high-dimensional probabilistic ODE solvers (Krämer et al., 2022), where a Kronecker-structured (i.e. separable) prior across state dimensions is coupled through conditioning on ODE residuals.

### 3.2. Sparse Information Form Inference

Inference in our framework repeatedly applies Equation (5). Naively, this involves $\mathcal{O}(N^2)$ memory and $\mathcal{O}(N^3)$ compute. Recently, approaches based on sparse precision matrices have successfully alleviated this issue (Chen et al., 2025; Weiland et al., 2025).

Assume we discretize $T$ into $\tilde{T} := [t_1, \ldots, t_{N_t}]$ and use the same spatial state $s$ across all time points. Then from Equation (13) the precision matrix of the joint distribution across all spacetime components factorizes into

$$\boldsymbol{Q}_{\tilde{T}, s} = \boldsymbol{Q}_{\tilde{T}} \otimes \boldsymbol{Q}_s, \quad (14)$$

with temporal precision $\boldsymbol{Q}_{\tilde{T}}$ and spatial precision $\boldsymbol{Q}_s$. $\boldsymbol{Q}_{\tilde{T}}$ is naturally sparse since the IWP is Markovian (Appendix A.1). Section 3.3 covers a sparse approximation to $\boldsymbol{Q}_s$.

Once we have a sparse approximation to Equation (14), we use sparse Cholesky factorizations for mean updates, sampling and to compute marginal variances. Memory and compute requirements depend on the sparsity pattern but generally scale much more favorably than dense equivalents. Weiland et al. (2025) discuss this inference scheme in detail.

### 3.3. Sparse Cholesky Approximation

Approximating $\boldsymbol{Q}_s$ with a sparse matrix $\tilde{\boldsymbol{Q}}_s$ is known in the literature as a Vecchia approximation (Katzfuss & Guinness,

---

**Algorithm 1** Joint Spatial Vecchia Approximation

**Input:** Kernel $k_{\text{space}}$, functionals $\{\ell_i\}$, sparsity $\rho$
Classify functionals by type (integral, evaluation, derivative)
Order within each type using maximin ordering
Concatenate: integrals $\rightarrow$ evaluations $\rightarrow$ derivatives
Compute sparsity pattern $S$ from ordering and $\rho$
Compute sparse Cholesky factor $\tilde{\boldsymbol{L}}$ via KL minimization
**Output:** Sparse precision matrix $\tilde{\boldsymbol{Q}}_s = \tilde{\boldsymbol{L}}\tilde{\boldsymbol{L}}^\top$

---

2021). Schäfer et al. (2021) frame the problem as finding a sparse Cholesky factor $\tilde{\boldsymbol{L}}$ such that the Kullback-Leibler (KL) divergence between $\mathcal{N}(\boldsymbol{0}, \boldsymbol{Q}_s^{-1})$ and $\mathcal{N}(\boldsymbol{0}, \tilde{\boldsymbol{L}}^{-\top}\tilde{\boldsymbol{L}}^{-1})$ is minimal. They find a closed-form expression for the optimal $\tilde{\boldsymbol{L}}$, where each column can be computed in parallel at cost that depends cubically on the sparsity pattern of the column. Thus, there is an inherent tradeoff for the sparsity pattern between accuracy and computational efficiency.

To motivate this idea, one can show that for $\boldsymbol{x} \sim \mathcal{N}(\boldsymbol{0}, \boldsymbol{L}^{-\top}\boldsymbol{L}^{-1})$, it holds:

$$\mathbb{E}[x_i \mid x_{i+1}, \ldots, x_N] = \mu_i - \sum_{j > i} \frac{L_{ji}}{L_{ii}}(x_j - \mu_j) \quad (15)$$

The key insight is that the $L_{ji}$ ($j > i$) form regression coefficients. For strategic node orderings and kernels with suitable properties, this results in so-called **screening effects** (Stein, 2011). This means that close points "screen out" distant points, causing their regression coefficients (and thus the $L_{ji}$) in Equation (15) to be close to zero. In this setting, Vecchia approximations achieve high accuracy.

**Node ordering.** Achieving strong screening effects requires a good ordering of the nodes $x_i$. In a pure regression setting, Schäfer et al. (2021) propose an optimal reverse maximin ordering. In a setting that mixes point evaluations with derivative evaluations, Chen et al. (2025) observe that ordering point evaluations before derivative evaluations results in stronger screening effects. Following this intuition of ordering "coarse" nodes before "fine nodes", we propose to order integrals before point evaluations, and point evaluations before derivatives. Within each category, we employ a reverse maximin ordering. See Appendix A.3 for details and empirical validation.

Algorithm 1 summarizes our approach based on the KL-minimizing sparse Cholesky factorization (Schäfer et al., 2021). Their algorithm includes a parameter $\rho \in \mathbb{R}^+$ which tunes the sparsity, allowing for an explicit tradeoff between accuracy and computation time. As in the original work, we use supernodes to benefit from BLAS-3 and LAPACK calls. Additionally, due to the block diagonal structure in Equation (10), we may apply this approximation to each block instead of the entire matrix at once.

## 3.4. An Alternative Treatment of Time

Working with a big joint distribution over all spacetime observations via Equation (14) is principled, but expensive. For inference, we need to compute sparse Cholesky factors of updated precision matrices, which induce numerical fill-in that quickly renders this approach prohibitively expensive. In the following, we discuss alternative strategies.

**Filtering and smoothing.** Probabilistic ODE solvers (Tronarp et al., 2019) use Kalman filter and smoother methods to achieve linear-in-time runtime complexity. Unfortunately, the Kalman filter prediction step produces a dense precision matrix even from a sparse prior. As such, it again induces a prohibitive cubic scaling in the spatial dimension.

**Marginal Moment-Matching.** We propose an alternative that approximates the filtering distributions by matching the marginal moments.

Given a current state distribution $s_t \sim \mathcal{N}(\mu_t, \Sigma_t)$ and transition distribution $s_{t+1} \mid s_t \sim \mathcal{N}(As_t, \Sigma_\varepsilon)$, the joint distribution is $(s_t, s_{t+1})^\top \sim \mathcal{N}(\mu, \Sigma)$ with mean $\mu = (\mu_t, A\mu_t)^\top$ and

$$\Sigma^{-1} = \begin{pmatrix} \Sigma_t^{-1} + A^\top \Sigma_\varepsilon^{-1} A & -A^\top \Sigma_\varepsilon^{-1} \\ -\Sigma_\varepsilon^{-1} A & \Sigma_\varepsilon^{-1} \end{pmatrix}. \qquad (16)$$

In our setting, $\Sigma_t^{-1}$ is the current sparse precision matrix, and $A$ and $\Sigma_\varepsilon^{-1}$ result from the state-space representation of Equation (13) together with the sparse approximation $\tilde{Q}_s$. Thus, $\Sigma^{-1}$ is sparse as a composition of sparse matrices. This enables us to efficiently perform inference on this two-step joint distribution.

From the conditioned joint distribution, we extract the mean $\tilde{\mu}_{t+1}$ and marginal variances $\tilde{\sigma}_{t+1}^2$ of the next state $s_{t+1}$ via selected inversion (Lin et al., 2011). We then construct the approximate filtering distribution for the next time step as

$$s_{t+1} \sim \mathcal{N}\left(\tilde{\mu}_{t+1}, \left(\tilde{Q}_s + \mathrm{diag}(\tilde{\sigma}_{t+1}^{-2})\right)^{-1}\right). \qquad (17)$$

This preserves the marginal means exactly while combining the prior precision structure with the inverse marginal variances. We repeat this procedure for each time step. In practice, we use a Crank-Nicolson discretization for the FVM constraints (Appendix A.2); the single Gauss–Newton step used for nonlinear constraints there is validated empirically in Appendix F.

This approach is linear in time and leaves the spatial sparsity pattern unchanged. However, it only provides an approximation to the true filtering distributions; we give a formal error analysis in Appendix G.

## 3.5. Complexity

The sparse Cholesky approximation of Schäfer et al. (2021) achieves near-linear complexity: $\mathcal{O}(N \log(N/\varepsilon)^d)$ space and $\mathcal{O}(N \log(N/\varepsilon)^{2d})$ time for $\varepsilon$-accuracy in $d$ dimensions. However, this requires access to covariance entries, which are cheap for the prior (kernel evaluations) but expensive for the posterior (requires inverting the precision). Thus, we use the sparse Cholesky approximation to obtain a sparse *prior* precision, but subsequent conditioning and inference use standard sparse Cholesky factorization (Chen et al., 2008), where fill-in is governed by the sparsity pattern. For 2D grids with nested dissection ordering, this yields $\mathcal{O}(N^{3/2})$ factorization time and $\mathcal{O}(N \log N)$ memory (Lipton et al., 1979). We validate this scaling empirically in Appendix D on the source-identification problem of Section 4.1. We emphasize that the functional ordering of Algorithm 1 (integrals $\to$ evaluations $\to$ derivatives) and the nested-dissection ordering serve distinct purposes: the former is a node ordering for the KL-minimizing sparse approximation that *produces* $\tilde{Q}_s$, whereas nested dissection is a fill-reducing permutation for *factorizing* the resulting sparse matrix. The $\mathcal{O}(N^{3/2})$ bound depends on the bounded-degree planar structure of $\tilde{Q}_s$ on a 2D mesh, not on the functional ordering used to construct it. Time-stepping via marginal moment-matching (Section 3.4) is linear in $N_t$, giving overall complexity $\mathcal{O}(N_t \cdot N_s^{3/2})$ for 2D problems.

# 4. Experiments

We evaluate four questions: Can our method solve ill-posed inverse problems with meaningful uncertainty estimates? Does it match classical FVM accuracy on forward problems? What is its convergence order? Does it scale to complex systems?

Our implementation is in Julia.[1] Hardware details and additional experimental parameters are provided in Appendix B.

## 4.1. Source Identification in Advection-Diffusion

In this experiment, we model the transport of contaminants in groundwater: A contaminant enters an aquifer and is spread downstream via the flow of water. In steady-state, this may be modeled by the advection-diffusion equation

$$\nabla \cdot (\mathbf{v}c - D\nabla c) = s, \qquad (18)$$

where $c$ is the contaminant concentration, $s$ is the contaminant source field, $\mathbf{v}$ the velocity field, and $D$ the dispersion coefficient. The inverse problem in this setting is source identification: Given measurements of the concentration $c(x, y)$ at sparse monitoring wells, recover the source field

---

[1]Code: https://github.com/timweiland/GPFiniteVolume.jl

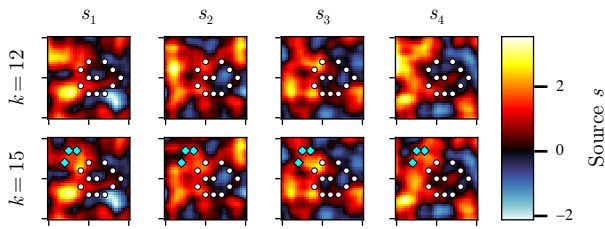

*Figure 2.* **Posterior samples adapt to observations.** Each column shows a posterior sample of the source field using the same random seed. Top row: 12 observations (○). Bottom row: 15 observations (3 additional shown as ◆). The added observations correctly inform the posterior about the weakness of the source in the top left corner, causing all samples to shift mass away from that region.

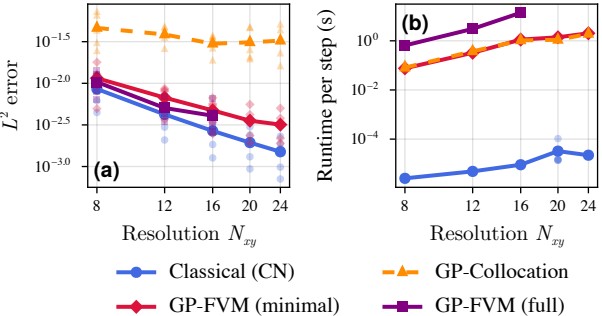

*Figure 3.* **Accuracy and runtime comparison on 2D Burgers equation.** (a) $L^2$ error vs grid size. Classical FVM achieves the best accuracy; GP-FVM (minimal) is within 2–3×; GP-Collocation is ∼10× worse. (b) Runtime per timestep. Classical methods are orders of magnitude faster, but GP-FVM provides uncertainty quantification. Faint markers show individual seeds; lines show means.

$s(x, y)$. This problem is ill-posed: Distinct source field configurations can yield similar downstream observations.

Figure 1 depicts the concrete setup. The ground-truth source $s$ (Figure 1e) consists of two bell-shaped regions where the contaminant enters. The contaminant then flows towards the right ($\boldsymbol{v} = (1\,0)^\top$) and diffuses ($D = 0.05$), resulting in a steady-state concentration $c$ (Figure 1a). This concentration is measured at 12 monitoring wells with additive Gaussian noise ($\sigma = 0.01$). For further details, refer to Appendix B.1.

Our method approaches this task through a joint distribution over concentration, source, and flux, as in Equation (10). As this is a steady-state problem (i.e. $\partial c/\partial t = 0$), we use purely spatial priors. The FVM constraints then automatically introduce coupling between the source and the concentration measurements. Posterior samples correspond to inferred possible forms of the source, while marginal means and variances yield a point estimate and confidence intervals.

**Baseline.** We compare against a physics-informed neural network (Raissi et al., 2019) baseline. It uses a dual-network architecture with separate MLPs for the concentration (4 layers × 128 units) and source field (2 layers × 64 units), trained jointly on PDE residuals, boundary conditions, and observation data. For a fair comparison, we follow state-of-the-art training practices (Wang et al., 2023), including Fourier feature embeddings and learning rate scheduling.

**Point estimates.** Figure 1 shows qualitative results of both methods. The posterior mean of our method correctly reconstructs the overall shape of both $c$ and $s$ (Figure 1c, g) and achieves a source RMSE of 0.44. By contrast, the PINN solution exhibits noticeable artifacts at a source RMSE of 0.76. **Our method recovers the source field in 2 seconds, while the PINN baseline requires approximately 5 minutes — a 150x speedup.**

**Marginal variances.** Our method provides uncertainty quantification on top of the point estimate — Figure 1d and Figure 1h show the marginal standard deviations. For both

$s$ and $c$, the uncertainty shrinks near the measurements. In addition, the uncertainty correctly reflects the fact that our measurements provide more information about $c$ — indeed, the uncertainty about $s$ is smeared due to the flow of water. The concentration uncertainty further collapses at the left boundary, which prescribes $c = 0$ (Figure 1d).

**Samples.** Our method's posterior not only provides marginal statistics, but also enables us to sample possible solution pairs of $(s, c)$. The top row of Figure 2 shows four samples of $s$ for our experiment. To further highlight that these samples are indeed physically meaningful, we repeat our experiment with three added observations and sample again using the same seeds. This is shown in the bottom row of Figure 2. The result is that for all samples, the top left corner indeed adapts and shifts the source mass, matching the ground-truth more closely.

**Takeaway.** The ill-posedness of source identification makes any estimate dependent on prior assumptions. Our method makes these explicit through the kernel; the PINN encodes them implicitly in the architecture and training procedure. In this case, a Matérn prior results in a better point estimate, and we *additionally* obtain confidence intervals and samples. We achieve all of this at a fraction of the cost of a PINN due to our proximity to classical numerical methods. We study the calibration of the posterior on this source identification problem in Appendix C, and Appendix E extends the same framework to a genuinely nonlinear inverse problem (Burgers source identification with a softplus link).

### 4.2. Accuracy and Runtime on Forward Problems

We benchmark our method on a forward problem: the 2D viscous Burgers' equation with bell-shaped initial conditions. We compare four methods: classical Crank-Nicolson

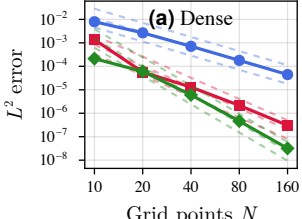
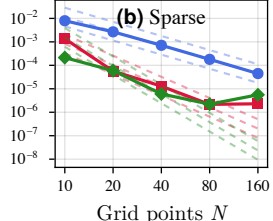

*Figure 4.* **Kernel smoothness dictates convergence order.** ● Matérn-3/2, ■ Matérn-5/2, ◆ Matérn-7/2. Dashed lines show theoretical rates. (a) Dense GP-FVM achieves higher-order convergence. (b) Sparse approximation ($\rho = 5$, 16% fill) preserves convergence for Matérn-3/2; smoother kernels plateau at fine resolutions.

FVM, a collocation-based sparse GP approach as in Chen et al. (2025), and our GP-FVM in both full and minimal state configurations. The minimal configuration uses 8 state components per cell (values, derivatives, integrals), while the full configuration additionally tracks flux quantities. For scalability in the time dimension, all GP approaches use our moment-matching technique. For each method, we vary the grid resolution $N \in \{8, 12, 16, 20, 24\}$ and average results over 5 random problem instances.

Figure 3 shows the results. Our GP-FVM (minimal) achieves accuracy within a factor of 2–3 of the classical method while providing uncertainty quantification. GP-Collocation, despite similar computational cost, performs an order of magnitude worse — highlighting the importance of conservative discretization. The full GP-FVM configuration achieves similar accuracy to the minimal variant but at substantially higher computational cost due to the larger state dimension. Full experimental specifications (problem setup, reference solver, kernel and lengthscale choices) are given in Appendix B.2.

### 4.3. Convergence with Kernel Smoothness

In classical FVM, higher-order polynomial reconstructions yield faster convergence rates. In our method, the analogous mechanism is kernel smoothness: Matérn-$\nu$ kernels with higher $\nu$ yield smoother GP samples and faster convergence.

Figure 4 demonstrates this on a 1D Poisson problem (Appendix B.3). Panel (a) shows dense GP-FVM achieving rates close to the theoretical rate for Matérn-based kernel interpolation $\mathcal{O}(h^{\nu+1/2})$ (Wendland, 2004). Panel (b) shows sparse GP-FVM with fixed $\rho = 5$ (16% fill at $N = 160$). Matérn-3/2 maintains its convergence rate perfectly. For smoother kernels, errors plateau or increase at fine resolutions, where screening effects weaken and sparse approximation struggles to capture long-range correlations. Note however that this occurs in a regime of already low $L^2$ error.

### 4.4. 2D Shallow Water Equations

To demonstrate scalability to complex nonlinear systems, we simulate tsunami propagation using the shallow water equations (Appendix B.4). This is a system of three coupled conservation laws governing water height $h$ and momentum $(hu, hv)$, with nonlinear flux terms that are quadratic in the conserved variables. The domain is $100\,\mathrm{km} \times 100\,\mathrm{km}$ with sloping bathymetry varying from $50\,\mathrm{m}$ depth at the shore to $500\,\mathrm{m}$ offshore, discretized on a $31 \times 31$ grid. The spatially varying bathymetry introduces source terms coupling the momentum equations to the bottom topography.

Figure 5 shows the evolution of surface elevation $\eta = h - b$ as a tsunami wave propagates toward shore. The method correctly captures wave propagation and shoaling—the steepening of waves as they enter shallow water—demonstrating physically plausible behavior without requiring problem-specific numerical treatment. This experiment shows that our framework extends naturally from scalar PDEs to complex systems of equations.

The joint state dimension per timestep is 28,566, comprising cell integrals, boundary values, and derivatives for all three conserved quantities. Our moment-matching scheme (Section 3.4) requires approximately 80 seconds per timestep, of which roughly 60 seconds is spent computing marginal variances via selected inversion. While slower than classical solvers, this cost enables uncertainty propagation through the nonlinear dynamics within a Bayesian framework.

## 5. Related Work

**GP-based PDE simulators.** Probabilistic numerical methods cast numerical computation as Bayesian inference, providing a coherent probabilistic treatment of numerical uncertainty (Cockayne, 2019). For PDEs, Pförtner et al. (2024) show that physics-informed GP regression strictly generalizes classical methods of weighted residuals, including collocation, finite volume, and Galerkin methods. Weiland et al. (2024) develop a probabilistic FVM using integral functionals, but their method is restricted to linear PDEs and their low-rank approach struggles to accurately quantify uncertainties for large-scale problems.

**Scalable inference.** Chen et al. (2025) apply sparse KL-minimizing Cholesky factorizations to solve nonlinear PDEs via collocation. Weiland et al. (2025) obtain scalability through the SPDE representation of Matérn priors in a Finite Element Method approach. By contrast, our work focuses on FVM, which is of central interest in computational fluid dynamics due to its conservative properties. By extending sparse KL-minimizing Cholesky to integral functionals, we frame FVM through a scalable Bayesian inference mechanism. Neither of these related works provides a practical answer to scalability along the time dimension.

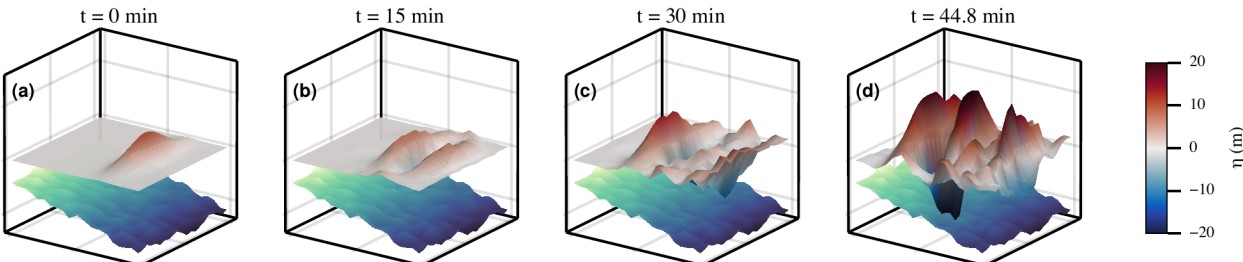

*Figure 5.* **Nonlinear shallow water simulation.** Evolution of the posterior mean over the surface elevation $\eta$ over a domain with spatially varying bathymetry (bottom surface). The method handles this system of three coupled conservation laws, demonstrating scalability to complex nonlinear problems.

## 6. Discussion

We frame the finite volume method as nonlinear conditioning under a structured GP prior. An extension of Vecchia approximations to integrals, as well as a novel moment-matching approach to time propagation, enables our method to circumvent the prohibitive cubic scaling of GPs. As a result, our method solves inverse problems quickly and under uncertainty and it scales to high-dimensional nonlinear simulations, while inheriting the accuracy and order-of-accuracy structure of classical FVM — with kernel smoothness playing the role of polynomial reconstruction order.

At the same time, our moment-matching approach incurs substantial fill-in in the temporal propagation step, which slows the underlying linear algebra and restricts practical scalability to 2D problems. The spatial sparse Cholesky scales well ($\mathcal{O}(N_s^{3/2})$ in 2D, $\mathcal{O}(N_s^2)$ in 3D), so the bottleneck lies in how time is handled rather than in the spatial machinery. This points toward future work on filtering approaches that preserve sparsity across time — for which our scalable spatial framework is precisely the substrate needed.

A second limitation concerns the expressiveness of the prior. For inverse problems with positivity-enforcing link functions (e.g. the softplus prior $s = \log(1 + e^g)$ on the source field), a stationary Matérn prior on the latent $g$ has constant marginal variance, and the saturating link compresses negative values of $g$ to near-zero $s$. In regions the posterior identifies as "no source present" ($g \ll 0$), the saturating link collapses the marginal variance of $s$, so the model underestimates its uncertainty about whether a weak source might be present. This is the locally overconfident source posterior visible where the softplus saturates in our Burgers experiment (Appendix E). Non-stationary kernels and sparsity-promoting priors (e.g. horseshoe-type) would relax this constraint; both remain compatible with the sparse Cholesky framework underlying our method.

Finally, our method relies on spatial Matérn priors and hyperrectangular FVM cells for closed-form computations of kernel integrals. Numerical integration would likely lift these restrictions, though we leave a thorough evaluation to future work.

**Scope and outlook.** Our contribution is twofold: a scalable methodology for FVM-based probabilistic inference under structured GP priors, and empirical evidence — forward accuracy on benchmark PDEs, $\mathcal{O}(N_s^{3/2})$ runtime scaling, and posterior coverage ranging from near-nominal to conservative in a calibration study — that the resulting posteriors are quantitatively meaningful. The Laplace approximation underlying our nonlinear inversion is of course only Gaussian-around-the-MAP; where richer non-Gaussian uncertainty is required, our scalable posterior provides a natural starting point and preconditioner for downstream samplers (e.g. as the initial state and mass matrix for Hamiltonian Monte Carlo, or as a proposal distribution), and therefore complements rather than competes with Markov chain Monte Carlo-based approaches. Beyond the engineering value of accelerated inference, the broader case for probabilistic numerical methods is that they place numerical and inferential uncertainty on a single footing: the same posterior that quantifies measurement noise and prior assumptions also quantifies discretization uncertainty, enabling coherent fusion of sparse data with mechanistic knowledge. Making this combination computationally practical for nonlinear conservation laws — the dominant PDE class in the sciences — is what our method delivers.

## Acknowledgements

The authors gratefully acknowledge co-funding by the European Union (ERC, ANUBIS, 101123955). Views and opinions expressed are however those of the author(s) only and do not necessarily reflect those of the European Union or the European Research Council. Neither the European Union nor the granting authority can be held responsible for them. PH is supported by the DFG through Project HE 7114/6-1 in SPP2298/2. PH is a member of the Machine Learning Cluster of Excellence, funded by the Deutsche Forschungsgemeinschaft (DFG, German Research Founda-

tion) under Germany's Excellence Strategy – EXC number 2064/1 – Project number 390727645. The authors also gratefully acknowledge the German Federal Ministry of Education and Research (BMBF) through the Tübingen AI Center (FKZ:01IS18039A); and funds from the Ministry of Science, Research and Arts of the State of Baden-Württemberg. The authors thank the International Max Planck Research School for Intelligent Systems (IMPRS-IS) for supporting TW.

## Impact Statement

This paper develops scalable Bayesian inference for nonlinear conservation laws, with the explicit goal of providing uncertainty quantification alongside numerical PDE solutions. The intended downstream uses include scientific decision-making under uncertainty — environmental monitoring, geophysical inversion, contaminant transport, and similar settings. A specific risk we wish to flag: the calibration of our posterior depends on prior choices (e.g. the lengthscale, smoothness, and stationarity of the spatial kernel) that encode modeling assumptions which may not hold for a given real-world problem. Our calibration study in Appendix C demonstrates broadly reasonable calibration — near-nominal to conservative — on a synthetic benchmark, but we caution against treating posterior uncertainty as automatically trustworthy in high-stakes settings (e.g. safety-critical or regulatory applications) without independent validation. We view the method as one source of evidence to be combined with domain expertise, sensitivity analyses, and corroborating data.

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

# A. Theoretical Details

## A.1. Integrated Wiener Process and Temporal Precision

The Integrated Wiener Process (IWP) of order $q$ is a Gauss-Markov process whose $q$-th derivative is Brownian motion. This Markov property induces a block tridiagonal structure in the joint precision matrix over discrete time points.

**State-space formulation.** The IWP($q$) can be written as a linear stochastic differential equation (SDE). Define the state vector $\boldsymbol{x}(t) := [u(t), u'(t), \ldots, u^{(q-1)}(t)]^\top \in \mathbb{R}^q$, collecting the function value and its first $q - 1$ derivatives. The SDE is

$$d\boldsymbol{x}(t) = \boldsymbol{F}\boldsymbol{x}(t)\, dt + \boldsymbol{B}\, dW(t), \tag{19}$$

where $W(t)$ is a Wiener process, $\boldsymbol{F} \in \mathbb{R}^{q \times q}$ is the shift matrix with ones on the superdiagonal and zeros elsewhere, and $\boldsymbol{B} = [0, \ldots, 0, \sigma]^\top \in \mathbb{R}^q$ drives only the highest derivative.

**Discretization.** Discretizing Equation (19) at time points $t_1, \ldots, t_{N_t}$ with step size $h = t_{i+1} - t_i$ yields the transition

$$\boldsymbol{x}_{i+1} = \boldsymbol{A}\boldsymbol{x}_i + \boldsymbol{\eta}_i, \quad \boldsymbol{\eta}_i \sim \mathcal{N}(\boldsymbol{0}, \boldsymbol{\Sigma}_\eta), \tag{20}$$

where $\boldsymbol{A} = \exp(\boldsymbol{F}h)$ is the state transition matrix. We have the closed forms (Schober et al., 2019):

$$A_{ij} = \frac{h^{j-i}}{(j-i)!} \quad \text{for } j \geq i, \quad A_{ij} = 0 \quad \text{otherwise,} \tag{21}$$

$$(\Sigma_\eta)_{ij} = \frac{\sigma^2 h^{2q+1-i-j}}{(q-i)!(q-j)!(2q+1-i-j)}. \tag{22}$$

**Block tridiagonal precision.** The Markov property in Equation (20) implies that the joint distribution $p(\boldsymbol{x}_1, \ldots, \boldsymbol{x}_{N_t})$ factorizes as

$$p(\boldsymbol{x}_1, \ldots, \boldsymbol{x}_{N_t}) = p(\boldsymbol{x}_1) \prod_{i=1}^{N_t-1} p(\boldsymbol{x}_{i+1} \mid \boldsymbol{x}_i). \tag{23}$$

Each transition density is Gaussian:

$$p(\boldsymbol{x}_{i+1} \mid \boldsymbol{x}_i) \propto \exp\left(-\tfrac{1}{2}(\boldsymbol{x}_{i+1} - \boldsymbol{A}\boldsymbol{x}_i)^\top \boldsymbol{\Pi}(\boldsymbol{x}_{i+1} - \boldsymbol{A}\boldsymbol{x}_i)\right), \tag{24}$$

where $\boldsymbol{\Pi} := \boldsymbol{\Sigma}_\eta^{-1}$ is the precision of the process noise. Expanding and collecting quadratic terms across all time steps, the joint precision matrix takes the block tridiagonal form

$$\boldsymbol{Q}_{\tilde{T}} = \begin{bmatrix} \boldsymbol{\Pi}_0 + \boldsymbol{A}^\top\boldsymbol{\Pi}\boldsymbol{A} & -\boldsymbol{A}^\top\boldsymbol{\Pi} & & \\ -\boldsymbol{\Pi}\boldsymbol{A} & \boldsymbol{\Pi} + \boldsymbol{A}^\top\boldsymbol{\Pi}\boldsymbol{A} & -\boldsymbol{A}^\top\boldsymbol{\Pi} & \\ & \ddots & \ddots & \ddots \\ & & -\boldsymbol{\Pi}\boldsymbol{A} & \boldsymbol{\Pi} \end{bmatrix}, \tag{25}$$

where $\boldsymbol{\Pi}_0$ is the precision of the initial state $\boldsymbol{x}_1$. Each block is $q \times q$, so for $N_t$ time steps the full matrix is $qN_t \times qN_t$ with only $\mathcal{O}(q^2 N_t)$ nonzero entries. This sparsity is what enables efficient temporal inference.

## A.2. Crank-Nicolson Time Discretization

For time-dependent problems, we enforce the FVM constraints using Crank-Nicolson discretization, which is second-order accurate in time and unconditionally stable for linear advection-diffusion problems.

**Two-step joint distribution.** At each time step, we form the joint distribution over consecutive states $(\boldsymbol{s}_{t-1}, \boldsymbol{s}_t)$ as described in Section 3.4. This joint distribution incorporates both the temporal prior (IWP transition) and the current spatial precision.

**Crank-Nicolson constraint.** The FVM constraint Equation (8) is enforced by averaging all quantities at consecutive time levels:

$$\frac{1}{2}\left(\dot{\bar{u}}_i^{(t-1)} + \dot{\bar{u}}_i^{(t)}\right) + \frac{1}{2}\left(\mathcal{F}_i^{(t-1)} + \mathcal{F}_i^{(t)}\right) = \frac{1}{2}\left(s_i^{(t-1)} + s_i^{(t)}\right), \tag{26}$$

where $\dot{\bar{u}}_i$ denotes the time derivative of the cell integral (which is explicitly available in the IWP state), $\mathcal{F}_i$ the net flux through cell boundaries, and $s_i$ the source integral.

**Conditioning.** We enforce this constraint on the joint distribution $(\boldsymbol{s}_{t-1}, \boldsymbol{s}_t)$ via the nonlinear conditioning procedure in Section 2.3. For linear PDEs, this reduces to a single linear constraint. For nonlinear PDEs, inspired by the Extended Kalman Filter (EKF), we take a *single* Gauss-Newton step rather than iterating to convergence. This is justified by the small time step assumption: with sufficiently small $\Delta t$, the nonlinearity within one step is mild, making a single linearization accurate. Any residual linearization error is corrected at subsequent time steps rather than accumulating, and the computational savings are substantial; we validate this empirically in Appendix F. After conditioning, we extract the marginal distribution of $\boldsymbol{s}_t$ and propagate to the next step.

**Laplace approximation error.** The Laplace approximation used in the per-step conditioning is exact for linear constraints and approximate for nonlinear ones. Combining the averaged conservation law (26) with the cell-wise PDE $\dot{\bar{u}}_i = s_i - \mathcal{F}_i$ and rearranging implicitly for $\boldsymbol{s}_t$, the constraint function has the structure

$$g(\boldsymbol{s}_t) = \boldsymbol{s}_t - \tfrac{\Delta t}{2}\, f(\boldsymbol{s}_t) - (\text{known from } \boldsymbol{s}_{t-1}), \tag{27}$$

i.e. identity plus an $O(\Delta t)$ nonlinear term. A short computation shows that all derivatives of $g$ beyond the first inherit the $\Delta t/2$ prefactor: $D^k g = O(\Delta t)$ for $k \geq 2$, whenever the flux $f$ is sufficiently smooth. Heuristically, a perturbative (Edgeworth-style) expansion of the negative log-posterior around the MAP then gives the following per-step rate.

*Remark* A.1 (Heuristic per-step Laplace rate). Suppose the flux $f$ in (27) is sufficiently smooth in a neighborhood of the MAP, the prior precision is bounded above and below, and the virtual-observation noise $\sigma$ is held fixed. A perturbative argument (informally summarized below) then indicates that the per-step KL divergence between the true posterior $p_t$ and the Laplace approximation $\hat{p}_t$ scales as

$$\mathrm{KL}(p_t \,\|\, \hat{p}_t) = O(\Delta t^2). \tag{28}$$

A rigorous treatment, tracking the dependence on the local Hessian condition number and on higher-order remainders, is left to future work.

*Remark* A.2 (Informal justification). The standardized higher-order corrections to a Laplace approximation scale with the third and higher derivatives of the negative log-posterior, evaluated at the MAP and rescaled by the local curvature. Since the negative log-likelihood under Gaussian virtual observations is $\frac{1}{2\sigma^2}\|g\|^2$ and the prior contributes only quadratic terms, all derivatives of the negative log-posterior of order $k \geq 3$ inherit the $\Delta t$ prefactor from $g$. The standardized non-Gaussian corrections are therefore $O(\Delta t)$ and the resulting KL error is $O(\Delta t^2)$. This is the Bayesian analogue of the standard observation that Crank-Nicolson is second-order accurate when the flux is smooth.

*Remark* A.3 (Limitation: shocks). The argument requires $f$ to have bounded derivatives near the MAP. For shock-dominated problems where the flux develops near-discontinuities, the higher-order derivatives blow up and the Laplace approximation can fail catastrophically. The same limitation applies to classical Crank-Nicolson schemes, which are known to require shock-capturing modifications in such regimes; addressing this within our framework is left to future work.

### A.3. Node Ordering for Mixed Functionals

When the state vector contains multiple types of linear functionals (point evaluations, derivatives, integrals), the sparse Cholesky approximation quality depends on how these blocks are ordered. We propose ordering functionals from finest to coarsest scale, placing integrals in the rightmost columns of the Cholesky factor.

**Functional classification.** We classify linear functionals into a hierarchy based on the spatial scale they capture:

- **Integrals** (coarsest): Cell averages $\iint f\, dx\, dy$ capture global, smoothed behavior
- **Face integrals**: Mixed tensor products such as $\int f(x, \cdot)\, dy$ at fixed $x$
- **Evaluations**: Point values $f(x_i)$ capture local behavior
- **Derivatives** (finest): Local slopes $\partial f/\partial x$ capture fine-scale variation

Our Julia implementation automatically classifies linear functionals via multiple dispatch on their types.

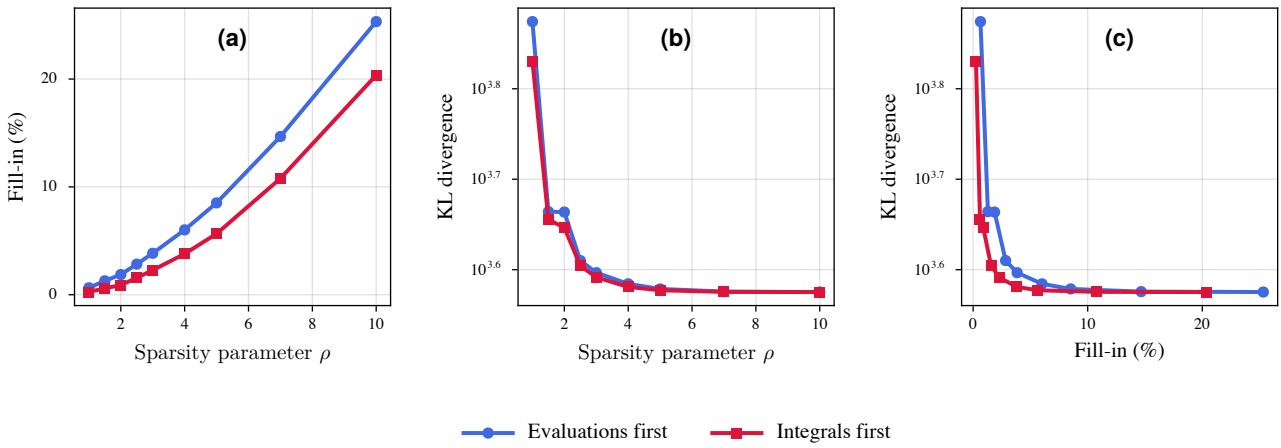

*Figure 6.* **Ordering comparison for sparse Cholesky approximation.** (a) Fill-in percentage vs sparsity parameter $\rho$. (b) KL divergence vs $\rho$. (c) Pareto frontier showing KL divergence vs fill-in. The "integrals first" ordering (red) achieves lower error at every sparsity level, demonstrating Pareto dominance over "evaluations first" (blue).

**Ordering algorithm.** We order functional blocks from finest to coarsest (derivatives $\to$ evaluations $\to$ integrals), placing the coarsest observations in the rightmost columns of the Cholesky factor. Within each block, we apply the reverse maximin ordering of Schäfer et al. (2021).

**Why integrals should be coarsest.** In the sparse Cholesky factorization, each row $i$ conditions on a subset of later rows $j > i$. The screening effect (Schäfer et al., 2021) implies that coarse-scale measurements effectively "screen out" correlations, reducing the conditioning set sizes needed for accurate approximation.

Integrals are smoothing operators that average over spatial regions, making them natural coarse-scale measurements. When integrals occupy the rightmost (coarsest) positions, all finer-scale functionals to their left benefit from this strong screening effect, requiring smaller conditioning sets to achieve the same approximation quality.

Additionally, there is a geometric asymmetry: point evaluations are more densely packed than integral cells covering the same domain. When integrals (in the middle) condition on evaluations (coarsest), each integral's neighborhood contains many evaluation points, leading to large conditioning sets. Conversely, when evaluations condition on integrals, each evaluation finds fewer integral neighbors. This asymmetry means that for the same sparsity parameter $\rho$, placing integrals coarsest yields lower fill-in.

**Empirical validation.** We compare two orderings on a 2D test problem: a $35 \times 35$ grid with point evaluations, derivative sums ($\partial_x + \partial_y$), and cell integrals under a Matérn-3/2 kernel. For each ordering, we sweep the sparsity parameter $\rho \in \{1, 1.5, 2, \ldots, 10\}$ and measure the KL divergence from the exact posterior.

Figure 6 shows that the "integrals first" ordering Pareto-dominates "evaluations first": for comparable fill-in levels, it achieves lower approximation error. This validates placing integrals as the coarsest block: both the screening effect and the geometric asymmetry favor this ordering.

**Visualizing the exact Cholesky factor.** Figure 7 shows the exact precision Cholesky factor $\boldsymbol{L}$ (where $\boldsymbol{\Theta} = \boldsymbol{L}\boldsymbol{L}^\top$) for a smaller $12 \times 12$ grid. The color scale shows $\log_{10} |L_{ij}|$, revealing that many remote entries are already close to zero in the exact factorization. This decay justifies the sparse approximation: entries below a threshold can be dropped with minimal loss in accuracy. The dashed lines mark boundaries between functional blocks (derivatives, evaluations and integrals).

## B. Experimental Details

All GP-based experiments were run on an Apple M1 Max with 32 GB RAM. The PINN baseline was run on an NVIDIA GeForce RTX 2080 Ti.

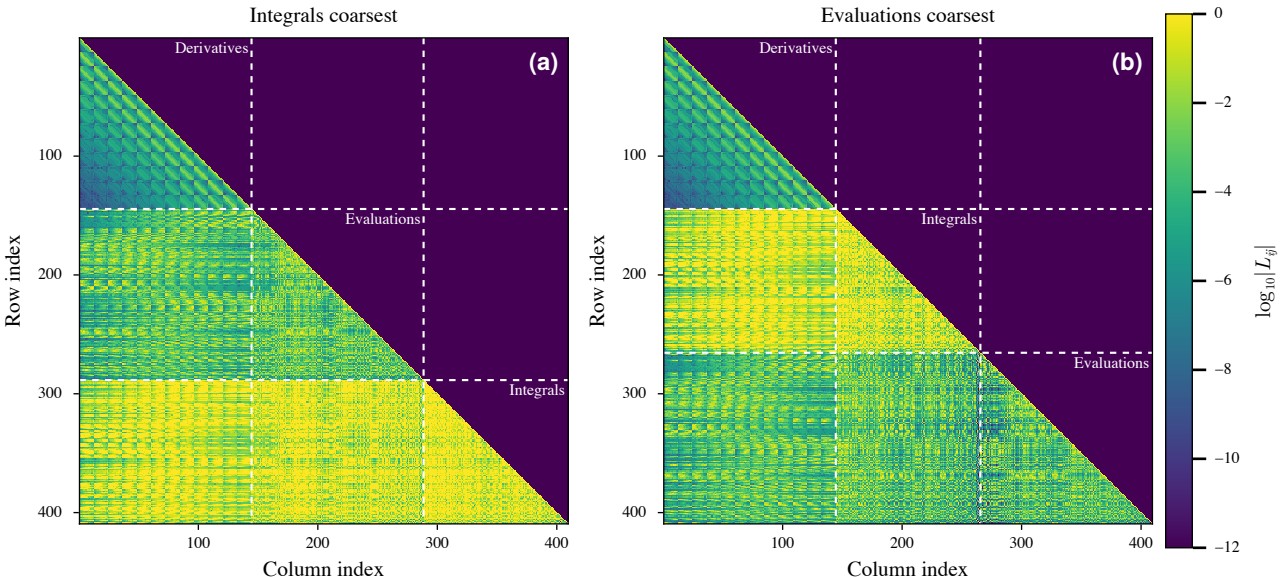

*Figure 7.* **Exact precision Cholesky factor for 2D mixed functionals.** Log-magnitude of entries in the lower triangular factor $\boldsymbol{L}$ for a $12 \times 12$ grid ($409 \times 409$ matrix). (a) Integrals-coarsest ordering places integrals in the bottom-right block. (b) Evaluations-coarsest ordering. Both show rapid decay of remote entries (dark blue regions), justifying sparse approximation. Dashed lines indicate block boundaries.

Our Julia implementation builds on `GaussianMarkovRandomFields.jl` (Weiland, 2025) for information form inference. This in turn builds on sparse linear algebra routines from SuiteSparse, specifically CHOLMOD (Chen et al., 2008). The sparse Jacobians required for Gauss-Newton steps are computed via sparse automatic differentiation, as in Hill & Dalle (2025).

### B.1. Source Identification in Advection-Diffusion

This section provides implementation details for the source identification experiment in Section 4.1.

**Problem specification.** The domain is the unit square $[0, 1]^2$ discretized on a uniform $36 \times 36$ node grid ($35 \times 35$ cells). The ground-truth source field consists of two Gaussian sources:

$$s_1(x, y) = 4.0 \cdot \exp\left(-\frac{(x - 0.25)^2 + (y - 0.3)^2}{2 \cdot 0.10^2}\right), \tag{29}$$

$$s_2(x, y) = 3.0 \cdot \exp\left(-\frac{(x - 0.35)^2 + (y - 0.7)^2}{2 \cdot 0.12^2}\right). \tag{30}$$

The physics parameters are velocity $\boldsymbol{v} = (1, 0)^\top$ and diffusion coefficient $D = 0.05$, yielding a Péclet number of $\mathrm{Pe} = v_x L / D = 20$ (advection-dominated). Boundary conditions are: Dirichlet $c = 0$ at the left boundary (clean inflow), homogeneous Neumann at the right boundary (free outflow), and homogeneous Neumann at the top and bottom boundaries (impermeable walls).

Ground-truth concentration data is generated via a forward FVM solve with upwind advection. Observations are taken at 12 monitoring well locations spread across the domain (Table 1), with additive Gaussian noise $\sigma = 0.01$.

**GP-FVM implementation.** We use Matérn-5/2 kernels ($\nu = 2$) with 2D tensor product structure. The lengthscale is set to $\ell = 5\Delta x \approx 0.14$, chosen to resolve approximately 5 grid cells. The sparsity parameter is $\rho = 2.0$ for the concentration field and $\rho = 4.0$ for the source field. The joint state vector comprises:

*Table 1.* Observation locations for the source identification experiment.

| $x$ | 0.50 | 0.50 | 0.60 | 0.70 | 0.70 | 0.80 | 0.80 | 0.90 | 0.40 | 0.40 | 0.50 | 0.60 |
|---|---|---|---|---|---|---|---|---|---|---|---|---|
| $y$ | 0.30 | 0.70 | 0.50 | 0.30 | 0.70 | 0.40 | 0.60 | 0.50 | 0.40 | 0.60 | 0.50 | 0.30 |

- **Concentration:** point evaluations $c(x_i, y_j)$, vertical face integrals $\int c \, dy$, horizontal face integrals $\int c \, dx$, derivative face integrals $\int \partial_x c \, dy$ and $\int \partial_y c \, dx$
- **Source:** point evaluations $s(x_i, y_j)$ and cell integrals $\iint_{\text{cell}} s \, dx \, dy$

All kernel integrals are computed in closed form using the Matérn polynomial structure (no quadrature). FVM and boundary condition constraints are enforced with precision $1/\sigma_{\text{constraint}}^2$ where $\sigma_{\text{constraint}} = 10^{-5}$.

**PINN baseline.** We implement the PINN baseline using the JAX-based `jaxpi` framework (Wang et al., 2023). The architecture uses two separate MLPs:

- **Concentration network:** 4 layers $\times$ 128 units, tanh activation, with Fourier feature embeddings (scale = 1.0, dimension = 128)
- **Source network:** 2 layers $\times$ 64 units, tanh activation, no Fourier features

We train with the Adam optimizer ($\beta_1 = 0.9$, $\beta_2 = 0.999$, initial learning rate $10^{-3}$) using exponential decay (rate 0.9 every 2000 steps) for 30,000 iterations. We use fixed loss weights: PDE residual = 1.0, boundary conditions = 1.0, observation data = 1000.0. We sample 4,096 interior collocation points and 100 boundary points per edge (400 total) at each iteration. All experiments use random seed 42. We also tested a larger source network (4 layers $\times$ 128 units with Fourier features), but found no improvement in reconstruction accuracy.

### B.2. 2D Burgers Benchmark

This section provides details for the accuracy and runtime comparison in Section 4.2.

**Problem specification.** We solve the 2D viscous Burgers equation in conservative form on the unit square $[0, 1]^2$ with periodic boundary conditions:

$$\frac{\partial u}{\partial t} + \frac{\partial}{\partial x}\left(\frac{1}{2}u^2\right) + \frac{\partial}{\partial y}\left(\frac{1}{2}u^2\right) = \nu\left(\frac{\partial^2 u}{\partial x^2} + \frac{\partial^2 u}{\partial y^2}\right), \tag{31}$$

with viscosity $\nu = 0.02$ and end time $T = 0.5$. Initial conditions are Gaussian bumps with randomly sampled parameters: width $\sigma \in [0.2, 0.3]$, amplitude $A \in [0.3, 0.6]$, and center $(x_0, y_0) \in [0.2, 0.5]^2$.

**Reference solution.** We compute ground-truth solutions using high-resolution classical FVM on a $128 \times 128$ grid with RK4 time integration and a stable CFL condition ($\Delta t = 0.25\Delta x^2$). Errors are computed as $L^2$ norm against this reference, interpolated to the test grid.

**Methods compared.**

- **Classical Crank-Nicolson FVM:** Standard implicit scheme with $N^2$ unknowns.
- **GP-Collocation:** Sparse GP approach following Chen et al. (2025), using point evaluations and derivatives. State dimension: $8N^2$.
- **GP-FVM (minimal):** Our method with point evaluations, derivatives, and cell integrals. State dimension: $8N^2$.
- **GP-FVM (full):** Adds flux-related quantities. State dimension: $26N^2$.

All methods use Crank-Nicolson time discretization (Appendix A.2). The GP methods additionally use Matérn-5/2 kernels with lengthscale $\ell = 0.3$ and our moment-matching scheme (Section 3.4) for temporal propagation.

**Experimental parameters.** Grid sizes: $N \in \{8, 12, 16, 20, 24\}$. Number of time steps scales with grid size: $n_{\text{steps}} \approx 1.5N$, so $\Delta t = T/n_{\text{steps}}$. We evaluate on five random problem instances.

**B.3. Poisson Convergence Study**

This section provides details for the convergence experiment in Section 4.3.

**Problem specification.** We solve the 1D Poisson equation on the unit interval:

$$-u''(x) = f(x), \quad x \in (0, 1), \quad u(0) = u(1) = 0. \tag{32}$$

The source is $f(x) = \sin(2\pi x)$, with exact solution $u(x) = \sin(2\pi x)/(4\pi^2)$.

**FVM formulation.** Integrating over cell $[x_i, x_{i+1}]$ and applying the fundamental theorem of calculus:

$$u'(x_i) - u'(x_{i+1}) = \int_{x_i}^{x_{i+1}} f(x) \, dx. \tag{33}$$

This links derivatives of $u$ at cell boundaries to integrals of the source $f$.

**GP-FVM implementation.** We place independent GP priors on $u$ and $f$, both with Matérn-$\nu$ kernels and lengthscale $\ell = 0.15$. The joint state for $u$ includes point evaluations $u(x_i)$, derivatives $u'(x_i)$, and cell integrals $\int u \, dx$. The joint state for $f$ includes point evaluations $f(x_i)$ and cell integrals $\int f \, dx$. We prescribe the source function $f$ at grid points, enforce the FVM constraints, and apply Dirichlet boundary conditions. All constraints use precision $\sigma_{\text{constraint}}^{-2}$ with $\sigma_{\text{constraint}} = 10^{-6}$.

**Experimental parameters.** Grid sizes: $N \in \{10, 20, 40, 80, 160\}$. Kernel smoothness: $\nu \in \{3/2, 5/2, 7/2\}$. For dense GP-FVM, we use $\rho = 60$ (essentially exact). For sparse GP-FVM, we use fixed $\rho = 5$, yielding approximately 16% fill-in at $N = 160$.

**B.4. Shallow Water Equations**

The nonlinear shallow water equations describe the evolution of water height $h$ and momentum $(hu, hv)$ over a domain with bathymetry $b(x, y)$:

$$\frac{\partial h}{\partial t} + \frac{\partial (hu)}{\partial x} + \frac{\partial (hv)}{\partial y} = 0, \tag{34}$$

$$\frac{\partial (hu)}{\partial t} + \frac{\partial}{\partial x}\left(\frac{(hu)^2}{h} + \frac{1}{2}gh^2\right) + \frac{\partial}{\partial y}\left(\frac{(hu)(hv)}{h}\right) = -gh\frac{\partial b}{\partial x}, \tag{35}$$

$$\frac{\partial (hv)}{\partial t} + \frac{\partial}{\partial x}\left(\frac{(hu)(hv)}{h}\right) + \frac{\partial}{\partial y}\left(\frac{(hv)^2}{h} + \frac{1}{2}gh^2\right) = -gh\frac{\partial b}{\partial y}, \tag{36}$$

where $g$ is gravitational acceleration. The right-hand side terms represent pressure forces from sloping bathymetry.

**Experimental setup.** We simulate tsunami propagation over a $100\,\text{km} \times 100\,\text{km}$ domain with linearly sloping bathymetry: depth varies from $50\,\text{m}$ at the shore ($x = 0$) to $500\,\text{m}$ offshore ($x = L_x$). The initial condition is a Gaussian perturbation of amplitude $10\,\text{m}$ centered at $(70\,\text{km}, 50\,\text{km})$, representing a tsunami source, with zero initial velocity. We do not include Coriolis forces or bottom friction.

## C. Calibration Study

Posterior standard deviation contracting near measurement locations is a generic property of GP regression; on its own, it does not establish that the posterior is calibrated, i.e. that credible intervals achieve their nominal coverage. We assess calibration empirically via a multi-instance study on the source identification problem of Section 4.1.

**Data generating process.** We generate 200 random source identification instances. Each instance has two Gaussian source bumps at random locations in $[0.2, 0.8]^2$ with random amplitudes in $[-3, 3]$ and widths in $[0.05, 0.15]$. For each instance, we solve the forward advection-diffusion PDE to obtain the true concentration, sample 10 noisy observations ($\sigma = 0.1$) at fixed locations, and run the GP-FVM solver on a $31 \times 31$ grid.

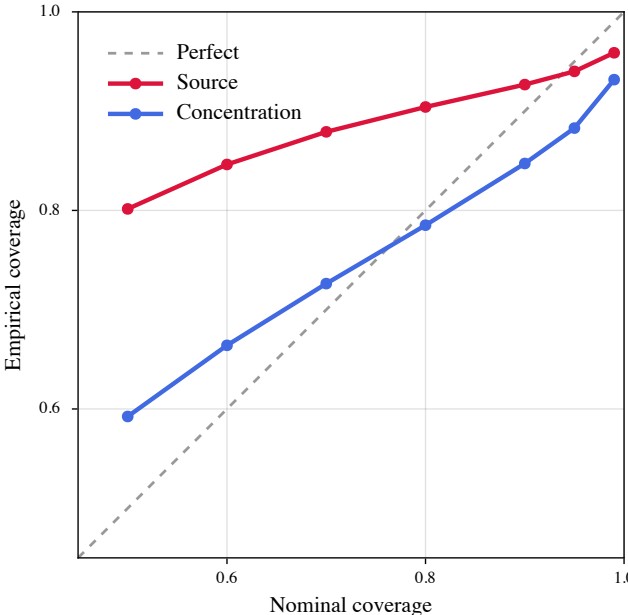

*Figure 8.* Calibration plot over 200 random two-source instances ($31 \times 31$ grid). Empirical coverage vs. nominal coverage level. The concentration field (blue) tracks the diagonal closely with mild deviations at the extremes; the source field (red) is broadly conservative and matches the diagonal near $95\%$.

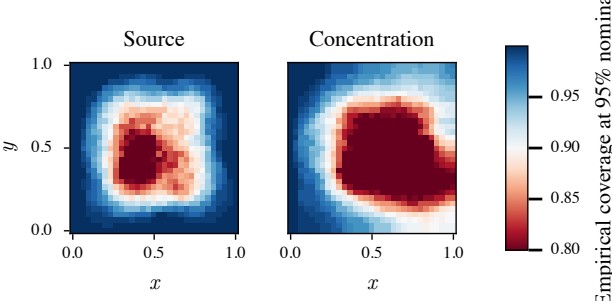

*Figure 9.* Spatial coverage map at 95% nominal level, averaged over 200 instances. Values near 0.95 (white) indicate well-calibrated regions. The source field is slightly overconfident in the interior where sources are active.

**Output scale calibration.** The GP prior's output scale $\sigma^2$ controls the overall magnitude of the posterior uncertainty. We calibrate it via a two-pass procedure: (i) run 10 calibration instances with an initial $\sigma^2$, compute the mean squared marginal $z$-score $\bar{z}^2 = \mathbb{E}[(f^\dagger - \mu)^2/\sigma_{\text{post}}^2]$; (ii) set $\sigma_{\text{cal}}^2 = \sigma_{\text{init}}^2 \cdot \bar{z}^2$ so that $z$-scores have unit variance on average. We calibrate the source and concentration fields independently, obtaining $\sigma_s^2 = 0.154$ and $\sigma_c^2 = 0.347$.

**Results.** Figure 8 shows the calibration plot (nominal vs. empirical coverage) across all 200 instances. The concentration field (blue) tracks the diagonal closely across most of the range, with mild deviations at the extremes: empirical coverage is somewhat above nominal at low levels (e.g. $59\%$ at $50\%$ nominal) and somewhat below at high levels ($88\%$ at $95\%$ nominal). The source field (red) is broadly conservative — empirical coverage substantially exceeds nominal across the low and middle range (e.g. $80\%$ at $50\%$ nominal, $93\%$ at $90\%$ nominal) — and crosses the diagonal near the $95\%$ level.

Figure 9 shows the spatial coverage map at the 95% level, averaged across all instances. The source field achieves near-nominal coverage at the domain boundaries (where the Dirichlet BC constrains the solution) and is slightly overconfident in the interior where the source bumps are located. This spatial pattern reflects the prior–truth mismatch: the stationary Matérn prior spreads uncertainty uniformly, while the actual error is concentrated at source locations.

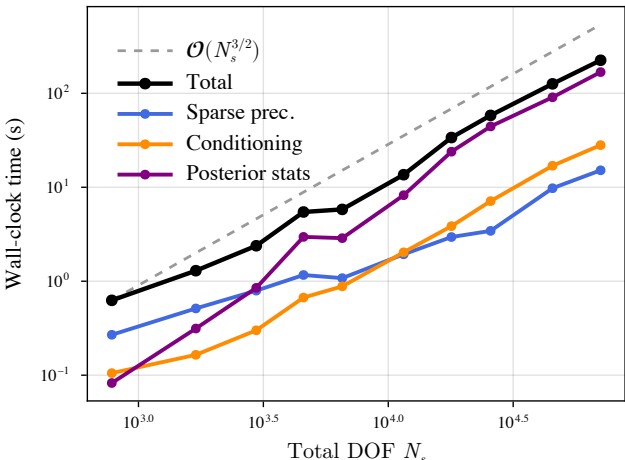

*Figure 10.* Wall-clock time vs. total degrees of freedom $N_s$ for the source identification problem. The dashed line shows the theoretical $\mathcal{O}(N_s^{3/2})$ scaling from nested-dissection sparse Cholesky. The total time (black) tracks this reference closely.

**Discussion.** The remaining miscalibration has two distinct sources:

1. **Prior–truth mismatch in the source field.** The stationary Matérn prior spreads uncertainty uniformly across the domain, while the actual reconstruction errors are concentrated at source locations. This produces the consistent over-coverage of the source curve at low and middle nominal levels: posterior credible intervals are wider than needed across most of the domain. Sparsity-promoting hierarchical GMRF priors (via local variance parameters with an appropriate hyperprior) or log-Gaussian priors for positivity would improve calibration while preserving the sparse Cholesky structure.

2. **Tail behavior of the concentration field.** The output-scale calibration matches the mean squared $z$-score to unity, ensuring the right average posterior spread but not the full shape of the empirical error distribution. The mild over-coverage at low nominal levels combined with mild under-coverage at high nominal levels is consistent with empirical errors that are heavier-tailed than the assumed Gaussian posterior. Heavier-tailed likelihood models or non-Gaussian posterior approximations could address this within the same sparse Cholesky framework.

Both limitations have concrete paths to improvement within the framework and do not require architectural changes.

## D. Scalability Study

To complement the asymptotic complexity analysis of Section 3.5, we report wall-clock timings and reconstruction errors for the source identification problem of Section 4.1 across a range of grid resolutions.

**Setup.** We run the 2D steady advection-diffusion source identification problem from Section 4.1 at grid sizes $N = 11$ to $N = 101$ ($N \times N$ grid points, corresponding to $N_s = 782$ to $N_s = 70{,}802$ total degrees of freedom). We use physics-motivated lengthscales ($\ell_c = 0.2 \approx \sqrt{D/v_x}$, $\ell_s = 0.12$) that are fixed across all grid sizes, and output scales $\sigma_s^2 = 0.154$, $\sigma_c^2 = 0.347$ taken from the calibration study (Appendix C). Timing is the best of 3 runs per grid size.

**Timing.** Figure 10 shows wall-clock time versus total DOF ($N_s$) on a log-log scale. The total time tracks the theoretical $\mathcal{O}(N_s^{3/2})$ reference line closely (empirical exponent $\approx 1.35$), confirming the complexity analysis in Section 3.5. The dominant cost at large $N_s$ is posterior statistics extraction (selected inversion on the sparse Cholesky factor). The method solves the inverse problem with $\sim 70{,}000$ DOF in under 4 minutes on a single CPU core.

**Error.** Table 2 shows that solution error is stable across grid sizes: source RMSE and concentration RMSE plateau beyond $N = 31$, consistent with the observation-limited regime (10 noisy measurements constrain the inverse problem, not the grid resolution). The source location error remains below one grid cell for all $N \geq 21$.

*Table 2.* Scalability results for the source identification problem. Error stabilizes beyond $N \approx 31$ (observation-limited); timing scales polynomially.

| $N$ | $N_s$ | Time (s) | Source RMSE | Conc. RMSE |
|---|---|---|---|---|
| 11 | 782 | 0.6 | 0.552 | 0.077 |
| 21 | 2,962 | 2.4 | 0.558 | 0.080 |
| 31 | 6,542 | 5.8 | 0.566 | 0.082 |
| 51 | 17,902 | 33.8 | 0.574 | 0.084 |
| 81 | 45,442 | 126.1 | 0.576 | 0.084 |
| 101 | 70,802 | 224.7 | 0.575 | 0.084 |

The error plateau confirms that the method does not degrade at larger grid sizes — the accuracy is limited by the observation information, not the discretization. The stable fill-in percentages (decreasing from 37.5% at $N = 11$ to 0.6% at $N = 101$ for the concentration field) demonstrate that the sparse Cholesky approximation becomes increasingly efficient at larger scales.

## E. Nonlinear Inverse Problem: Burgers Source Identification

We demonstrate our framework on a genuinely nonlinear inverse problem: inferring an unknown spatial source field $s(x, y) \geq 0$ from noisy observations of the solution $u(x, y, t)$ to the 2D viscous Burgers equation,

$$\frac{\partial u}{\partial t} + \frac{\partial}{\partial x}\left(\frac{u^2}{2}\right) + \frac{\partial}{\partial y}\left(\frac{u^2}{2}\right) = \nu \Delta u + s(x, y), \tag{37}$$

on $[0, 1]^2 \times [0, 0.5]$ with $\nu = 0.01$, homogeneous Dirichlet boundary conditions, and Gaussian bump initial condition. The source $s$ is constant in time and couples all time steps.

**Setup.** The source consists of two Gaussian bumps with amplitudes $\approx 1$ and $\approx 3.5$ and widths $\approx 0.12$–$0.14$. We observe $u$ at 50 random spatial locations at 5 out of 10 time steps (250 observations total, noise $\sigma = 0.05$). The source is never observed directly — it must be inferred entirely through the nonlinear PDE.

**Prior and solver.** We place a Matérn-5/2 GP prior on $u$ with IWP(1) temporal dynamics and a separate Matérn-5/2 GP prior on $g$, where $s = \text{softplus}(g) = \log(1 + e^g) \geq 0$. The softplus link enforces positivity while keeping bounded Jacobian entries (unlike $\exp$), which improves the quality of the Gauss–Newton approximation. The joint spacetime GMRF has $\sim 22{,}000$ degrees of freedom ($16 \times 16$ spatial grid, 11 time steps). The Gauss–Newton iteration converges in 14 iterations (Table 3), with Newton decrement dropping over 10 orders of magnitude.

**Results.** Figure 11 shows the solution at five time steps (IC, two unobserved, two observed). The GP-FVM posterior mean tracks the nonlinear advection–diffusion dynamics closely. The posterior standard deviation (bottom row) is small near observation locations (white dots) and larger between observed time steps, reflecting the expected information pattern.

Figure 12 shows the inferred source field. The posterior mean identifies both source locations and captures the dominant bump's amplitude. The posterior standard deviation (in $g$-space, interpretable as multiplicative uncertainty) is largest where the source is active, reflecting the difficulty of the inverse problem.

**Discussion.** This experiment demonstrates two properties of the framework: (i) it handles a genuinely nonlinear inverse problem (Burgers flux $F = u^2/2$), and (ii) the Gauss–Newton iteration converges reliably to a good solution in $\sim 10$–$14$ iterations. The softplus source prior demonstrates that nonlinear link functions (and the associated positivity constraints) integrate naturally into the GP-FVM framework — the same machinery handles both the PDE nonlinearity and the prior nonlinearity.

The remaining locally underestimated uncertainty in the source posterior (visible where the softplus is saturated) is discussed in detail in Section 6.

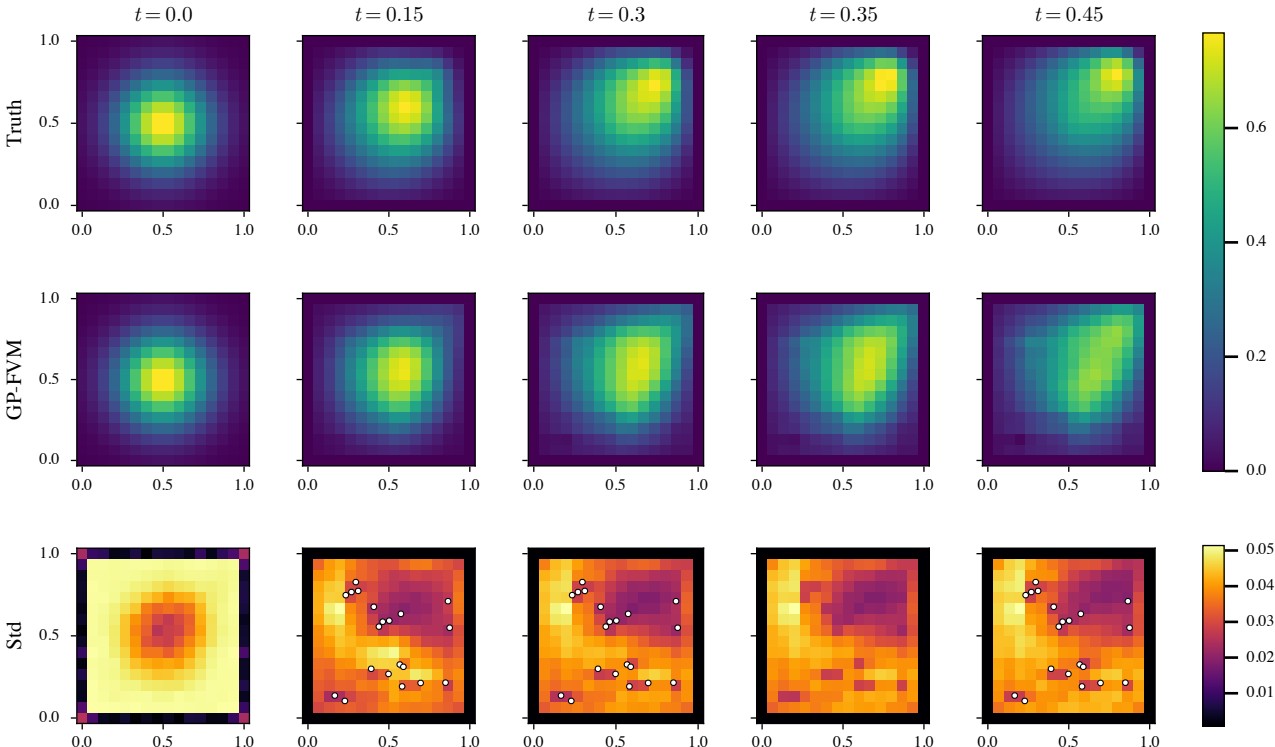

*Figure 11.* Nonlinear inverse problem: 2D Burgers source identification. **Top:** ground truth $u(x, y, t)$. **Middle:** GP-FVM posterior mean. **Bottom:** posterior standard deviation; white dots indicate observation locations. The method correctly captures the nonlinear advection to the upper right while jointly inferring the unknown source.

## F. Gauss–Newton Ablation

The sequential (EKF-style) solver of Section 3.4 performs a single Gauss–Newton step per time step. This is a strong assumption, which we validate empirically in two complementary settings: a forward-simulation ablation showing that additional iterations do not improve accuracy, and a convergence study confirming that the joint solver used for inverse problems requires multiple iterations but reaches them reliably.

**Ablation: sequential solver.** We run the 1D viscous Burgers forward problem with the EKF sequential solver, varying the number of Gauss–Newton iterations per time step from 1 (standard EKF) to 10. Figure 13 shows the result across 5 initial conditions (sine and Gaussian families) at $N = 100$ grid points: additional iterations provide **no measurable improvement** in solution accuracy ($< 0.1\%$ change) while adding $\sim 10\%$ computational overhead.

This validates the single-step assumption: in the small-$\Delta t$ regime, the per-step nonlinearity is a mild perturbation, and one linearization suffices. The Gauss–Newton iteration converges in a single step because the prior from the previous time step already provides an excellent linearization point.

**Convergence: joint solver.** For the nonlinear inverse problem (Appendix E), where we solve the full spacetime system jointly, multiple Gauss–Newton iterations *are* needed — the problem cannot be decomposed into small-$\Delta t$ steps. The iteration converges reliably in $\sim 10$–$14$ steps (Table 3), with the Newton decrement dropping over 10 orders of magnitude.

## G. Moment-Matching Error Analysis

We provide a theoretical analysis of the moment-matching approximation introduced in Section 3.4. We frame it within the assumed density filtering framework, derive a per-step projection error decomposition that explicitly accounts for the diagonal mismatch arising from the closed-form variance rule, and bound the accumulated KL divergence from the exact filter via a contraction argument. The analysis shows that the moment-matching error decomposes into an off-diagonal

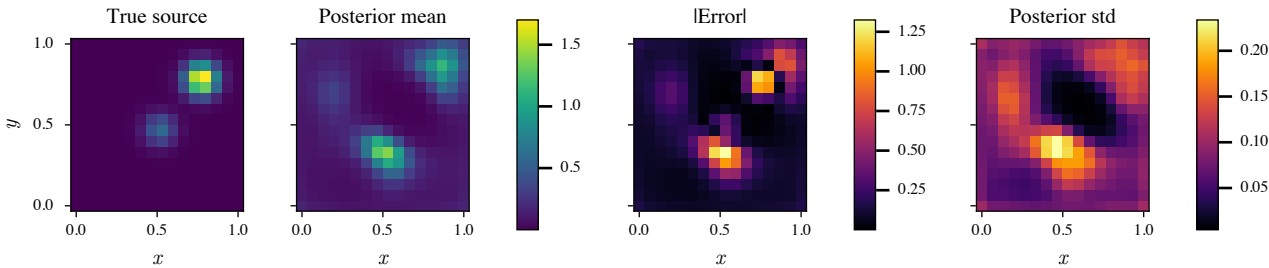

*Figure 12.* Inferred source field. Left to right: ground truth $s(x, y)$, posterior mean $\mathrm{softplus}(\hat{g})$, absolute error, and posterior std in $g$-space. Both source bumps are identified; the dominant source (amplitude $\approx 3.5$) is well-recovered.

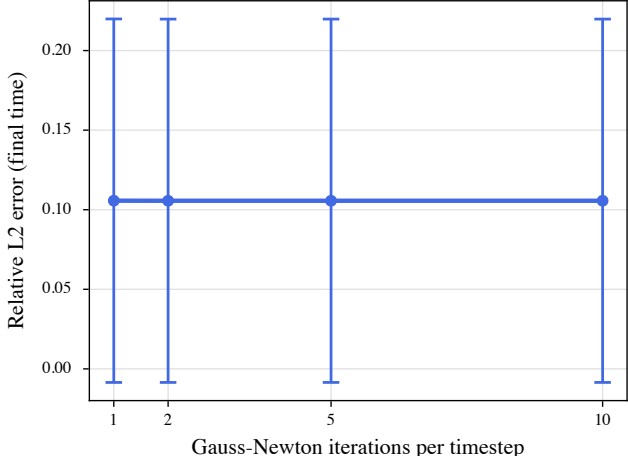

*Figure 13.* Gauss–Newton ablation for the EKF sequential solver on 1D Burgers ($N = 100$, 5 Gaussian ICs). Additional iterations beyond 1 provide no accuracy improvement.

contribution that vanishes as $\Delta t \to 0$ and a diagonal mismatch floor controlled by the spatial prior's conditioning.

### G.1. Setup and Notation

**Definition G.1** (Approximating family). Let $Q_s \succ 0$ be the sparse spatial prior precision. The approximating family is

$$\mathcal{F} := \left\{ \mathcal{N}\left(\mu, (Q_s + D)^{-1}\right) : \mu \in \mathbb{R}^N, \ D = \mathrm{diag}(d), \ d > 0 \right\}. \tag{38}$$

**Standing notation.**

- $\Sigma_t^+$: marginal filtering covariance of $x_t$ after conditioning at step $t$.

- $\Sigma_t^-$: marginal predictive covariance before conditioning.

- $A = A(\Delta t)$: IWP transition matrix; $\Sigma_\eta = \Sigma_\eta(\Delta t)$: process noise covariance.

- $\widetilde{\Sigma}_t = (Q_s + \widetilde{D}_t)^{-1}$: moment-matched covariance, with $\widetilde{D}_t = \mathrm{diag}(\sigma_t^{-2})$.

- $\sigma_{t,i}^2 := [\Sigma_t^+]_{ii}$: the true marginal variances after conditioning.

- $\Delta_t := G_t S_t^{-1} G_t^\top \succeq 0$: covariance removed by FVM conditioning.

- $\lambda_{\min}(\cdot), \lambda_{\max}(\cdot)$: smallest and largest eigenvalues.

*Table 3.* Gauss–Newton convergence for the Burgers source identification problem ($16 \times 16$ grid, 11 time steps, 22k DOF).

| Iter | Newton dec. | $\alpha$ |
|------|-------------|----------|
| 1 | $1.5 \times 10^6$ | 1.0 |
| 2 | $4.7 \times 10^4$ | 1.0 |
| 3 | $5.5 \times 10^2$ | 1.0 |
| 5 | $4.1 \times 10^1$ | 1.0 |
| 7 | $2.0 \times 10^{-1}$ | 0.31 |
| 10 | $1.1 \times 10^2$ | 0.86 |
| 14 | $6.4 \times 10^{-4}$ | 0.29 |

- $\kappa(M) := \lambda_{\max}(M)/\lambda_{\min}(M)$: condition number.

- $N$: state dimension; $N_s$: number of spatial cells; $m$: observation dimension per step.

**Key spectral parameters.**

$$\lambda_{\min}(Q_s) =: \alpha > 0, \qquad \lambda_{\max}(Q_s) =: \beta, \qquad \kappa_s := \beta/\alpha. \tag{39}$$

The filtering precision satisfies $Q_s + \widetilde{D}_t \succeq Q_s \succeq \alpha I$, giving the covariance upper bound

$$\widetilde{\Sigma}_t \preceq Q_s^{-1} \preceq \alpha^{-1} I. \tag{40}$$

We assume throughout that $\|\widetilde{D}_t\|_{\mathrm{op}} \leq \delta_{\max}$ uniformly in $t$ for some finite $\delta_{\max} > 0$; this is a mild filter-stability condition, satisfied whenever the moment-matched marginals are bounded away from zero. Under this assumption the covariance lower bound is

$$\widetilde{\Sigma}_t \succeq (\beta + \delta_{\max})^{-1} I. \tag{41}$$

### G.2. The Diagonal Mismatch

The algorithm stores $\sigma_{t,i}^2 := [\Sigma_t^+]_{ii}$ and sets $\widetilde{D}_t = \mathrm{diag}(1/\sigma_{t,i}^2)$. If $Q_s = 0$, then $\widetilde{\Sigma}_t = \widetilde{D}_t^{-1}$ and $[\widetilde{\Sigma}_t]_{ii} = \sigma_{t,i}^2$ exactly. But for $Q_s \neq 0$, the Schur complement formula gives

$$[(Q_s + D)^{-1}]_{ii} = \frac{1}{d_i + [Q_s]_{ii} - \mathbf{q}_i^\top (Q_{s,-i} + D_{-i})^{-1} \mathbf{q}_i}, \tag{42}$$

where $\mathbf{q}_i$ is the $i$-th row of $Q_s$ with the diagonal removed, and $Q_{s,-i}, D_{-i}$ are the corresponding submatrices. The correction term $c_i := \mathbf{q}_i^\top (Q_{s,-i} + D_{-i})^{-1} \mathbf{q}_i \geq 0$ makes the actual marginal variance differ from $1/(d_i + [Q_s]_{ii})$, shifting it away from the target $\sigma_{t,i}^2 = 1/d_i$.

**Lemma G.2** (Diagonal mismatch bound)**.** *Define the diagonal mismatch $\eta_{t,i} := [\widetilde{\Sigma}_t]_{ii} - \sigma_{t,i}^2$. Then:*

$$|\eta_{t,i}| \leq \frac{\beta^2}{\alpha^2\, d_{\min,t}^2} =: \bar{\eta}_t, \tag{43}$$

*where $d_{\min,t} := \min_i [\widetilde{D}_t]_{ii} = 1/\max_i \sigma_{t,i}^2$. Moreover, the Neumann expansion gives the leading-order expression*

$$\eta_{t,i} = -[Q_s]_{ii}\, \sigma_{t,i}^4 + O(\beta^2/d_{\min,t}^3). \tag{44}$$

*In particular, $\eta_{t,i} < 0$ to leading order: the closed-form rule undershoots the target variance.*

*Proof.* From (42) with $d_i = 1/\sigma_{t,i}^2$:

$$[\widetilde{\Sigma}_t]_{ii} = \frac{1}{1/\sigma_{t,i}^2 + [Q_s]_{ii} - c_i}, \qquad \eta_{t,i} = \frac{1}{1/\sigma_{t,i}^2 + [Q_s]_{ii} - c_i} - \sigma_{t,i}^2.$$

Since $Q_{s,-i} + D_{-i} \succeq \alpha I$, we have $0 \leq c_i \leq \|\mathbf{q}_i\|^2/\alpha \leq \beta^2/\alpha$ (using $\|\mathbf{q}_i\|^2 \leq \|Q_s\, e_i\|^2 \leq \|Q_s\|_{\mathrm{op}}^2 \leq \beta^2$). Let $r_i := [Q_s]_{ii} - c_i$. Then:

$$\eta_{t,i} = \frac{1}{1/\sigma_{t,i}^2 + r_i} - \sigma_{t,i}^2 = \frac{-r_i\,\sigma_{t,i}^2}{1/\sigma_{t,i}^2 + r_i} = \frac{-r_i\,\sigma_{t,i}^4}{1 + r_i\,\sigma_{t,i}^2}.$$

Since $r_i = [Q_s]_{ii} - c_i$ can be positive or negative in general, but $1/\sigma_{t,i}^2 + r_i > 0$ (the Schur complement is positive because $Q_s + D \succ 0$), we have:

$$|\eta_{t,i}| = \frac{|r_i|\,\sigma_{t,i}^4}{1 + r_i\,\sigma_{t,i}^2} \leq |r_i|\,\sigma_{t,i}^4 \leq (\beta + \beta^2/\alpha)\,\sigma_{t,i}^4,$$

and crudely $|\eta_{t,i}| \leq \beta^2 \sigma_{t,i}^4/\alpha^2 \leq \beta^2/(\alpha^2 d_{\min,t}^2)$ since $\sigma_{t,i}^4 \leq 1/d_{\min,t}^2$.

For the leading-order expression, expand $(Q_s + D)^{-1} = D^{-1} - D^{-1}Q_sD^{-1} + D^{-1}Q_sD^{-1}Q_sD^{-1} - \cdots$ (valid when $\|D^{-1}Q_s\|_{\mathrm{op}} < 1$, i.e., $d_{\min,t} > \beta$). The diagonal of the first correction is $-[D^{-1}Q_sD^{-1}]_{ii} = -[Q_s]_{ii}/d_i^2 = -[Q_s]_{ii}\,\sigma_{t,i}^4$, which is negative since $[Q_s]_{ii} > 0$. $\qquad \square$

*Remark G.3* (When is the mismatch small?). The mismatch is controlled by the ratio $\beta^2/(\alpha^2 d_{\min,t}^2)$. As the filter accumulates temporal information ($d_{\min,t}$ grows), $\bar\eta_t \to 0$: the spatial prior becomes a perturbation of the total precision. In the Riccati steady state with time step $\Delta t$, $d_{\min,t}$ stabilizes at a value determined by the balance between prediction and update, and $\bar\eta_t$ is $O(1)$ with respect to $\Delta t$ — it depends on the prior conditioning but not on the temporal resolution. During the initial transient ($d_{\min,t}$ small), the mismatch can be large, but these errors are exponentially forgotten by the contraction argument (Appendix G.6).

## G.3. Per-Step Update

The prediction and update follow the standard Kalman recursion, with the approximate filter propagating $\widetilde\Sigma_{t-1}$ (the moment-matched covariance) rather than $\Sigma_t^+$:

$$\Sigma_t^- = A\,\widetilde\Sigma_{t-1}\,A^\top + \Sigma_\eta, \tag{45}$$

$$\Sigma_t^+ = \Sigma_t^- - \Delta_t, \qquad \Delta_t := G_t S_t^{-1} G_t^\top \succeq 0. \tag{46}$$

## G.4. Per-Step Projection Error

The covariance error $E_t := \Sigma_t^+ - \widetilde\Sigma_t$ has two contributions: (a) off-diagonal discrepancy from projecting onto the sparse precision family; and (b) diagonal discrepancy from the approximate moment-matching rule.

**Lemma G.4** (Projection error). *Let $\hat p_t = \mathcal{N}(\mu_t, \Sigma_t^+)$ and $p_t = \mathcal{N}(\mu_t, \widetilde\Sigma_t) \in \mathcal{F}$. Then:*

(a) *The covariance error satisfies $\mathrm{diag}(E_t) = -\eta_t$ (the diagonal mismatch from Lemma G.2), so $E_t$ does* not *have zero diagonal.*

(b) *The Frobenius norm of the error decomposes as*

$$\|E_t\|_{\mathrm{F}}^2 = \|E_t^{\mathrm{off}}\|_{\mathrm{F}}^2 + \|\eta_t\|_2^2, \tag{47}$$

*where $E_t^{\mathrm{off}}$ denotes the off-diagonal part. The off-diagonal contribution satisfies $\|E_t^{\mathrm{off}}\|_{\mathrm{F}} \leq C_Q\|\Delta_t\|_{\mathrm{F}}$, where $C_Q$ depends on $\kappa_s$ and $\delta_{\max}$ (see proof).*

(c) *The projection error satisfies*

$$\varepsilon_t := \mathrm{KL}(\hat p_t\|p_t) \leq \frac{(\beta + \delta_{\max})^2}{2}\left(C_Q^2\,\|\Delta_t\|_{\mathrm{F}}^2 + \|\eta_t\|_2^2\right) + h.o.t. \tag{48}$$

*Proof.* **Part (a).** $[E_t]_{ii} = [\Sigma_t^+]_{ii} - [\widetilde\Sigma_t]_{ii} = \sigma_{t,i}^2 - (\sigma_{t,i}^2 + \eta_{t,i}) = -\eta_{t,i}$.

**Part (b).** Let $P = Q_s + D'$ approximate the predictive precision, so $\Sigma_t^- \approx P^{-1}$ (with $O(\Delta t)$ error from the prediction step). After the FVM update: $\Sigma_t^+ = P^{-1} - \Delta_t$. The moment-matching sets $\widetilde D_t = \mathrm{diag}(1/[\Sigma_t^+]_{ii})$ and forms $\widetilde\Sigma_t = (Q_s + \widetilde D_t)^{-1}$.

Write $\widetilde{D}_t = D' + \delta D$ for some diagonal perturbation $\delta D$. By the resolvent identity:

$$(Q_s + \widetilde{D}_t)^{-1} = P^{-1} - P^{-1}\,\delta D\, P^{-1} + O(\|\delta D\|_{\mathrm{op}}^2).$$

Thus $E_t = (P^{-1} - \Delta_t) - (P^{-1} - P^{-1}\delta D\, P^{-1}) + O(\|\delta D\|_{\mathrm{op}}^2) = P^{-1}\delta D\, P^{-1} - \Delta_t + \text{h.o.t.}$

The off-diagonal part of $E_t$ is $(P^{-1}\delta D\, P^{-1})^{\mathrm{off}} - \Delta_t^{\mathrm{off}}$. Since $\|P^{-1}\|_{\mathrm{op}} \le \alpha^{-1}$:

$$\|P^{-1}\delta D\, P^{-1}\|_{\mathrm{F}} \le \alpha^{-2}\|\delta D\|_{\mathrm{F}} = \alpha^{-2}\|\delta d\|_2.$$

The perturbation $\delta d$ is determined by the moment-matching condition $\mathrm{diag}(\widetilde{\Sigma}_t) = \mathrm{diag}(\Sigma_t^+)$, which couples $\delta d$ to both $\Delta_t$ (the update term) and the Schur-complement diagonal mismatch from Lemma G.2. Expanding to leading order in $\|\Delta_t\|_{\mathrm{op}}$, and absorbing the prediction-step approximation $P^{-1} \approx \Sigma_t^-$ together with the constants $\alpha, \beta$ into a single coefficient $C_Q = O(\alpha^{-2}\beta^2)$, we obtain the working bound

$$\|E_t^{\mathrm{off}}\|_{\mathrm{F}} \le C_Q\,\|\Delta_t\|_{\mathrm{F}}, \tag{49}$$

which we use in part (c) below. A fully rigorous derivation of $C_Q$ that tracks the prediction-step coupling and the diagonal mismatch jointly is beyond the scope of this analysis.

**Part (c).** For Gaussians with the same mean, $\mathrm{KL}(\hat{p}_t\|p_t) = \frac{1}{2}[\mathrm{tr}(\widetilde{\Sigma}_t^{-1}\Sigma_t^+) - N + \log\det(\widetilde{\Sigma}_t\,(\Sigma_t^+)^{-1})]$. Writing $\Sigma_t^+ = \widetilde{\Sigma}_t + E_t$ and $W := \widetilde{\Sigma}_t^{-1/2}E_t\,\widetilde{\Sigma}_t^{-1/2}$:

$$\mathrm{KL}(\hat{p}_t\|p_t) = \frac{1}{2}\sum_i \big[\lambda_i(W) - \log(1 + \lambda_i(W))\big] \le \frac{\|W\|_{\mathrm{F}}^2}{4(1 - \|W\|_{\mathrm{op}})} \tag{50}$$

for $\|W\|_{\mathrm{op}} < 1$, using $x - \log(1+x) \le x^2/(2(1-|x|))$.

Now $\|W\|_{\mathrm{F}} \le \|\widetilde{\Sigma}_t^{-1}\|_{\mathrm{op}}\|E_t\|_{\mathrm{F}} \le (\beta + \delta_{\max})\|E_t\|_{\mathrm{F}}$ by (41). For $\Delta t$ small enough that $\|W\|_{\mathrm{op}} \le 1/2$:

$$\varepsilon_t \le \frac{(\beta + \delta_{\max})^2}{2}\|E_t\|_{\mathrm{F}}^2 = \frac{(\beta + \delta_{\max})^2}{2}\big(\|E_t^{\mathrm{off}}\|_{\mathrm{F}}^2 + \|\eta_t\|_2^2\big).$$

Substituting the bound on $\|E_t^{\mathrm{off}}\|_{\mathrm{F}}$ gives (48). $\qquad\square$

*Remark* G.5 (Two error sources). The projection error has two additive contributions with distinct characters:

1. The *off-diagonal* term $C_Q^2\|\Delta_t\|_{\mathrm{F}}^2$ arises from replacing the full posterior correlation structure with the sparse prior structure. This is $O(\|\Delta_t\|_{\mathrm{F}}^2)$ and vanishes as $\Delta t \to 0$.

2. The *diagonal mismatch* term $\|\eta_t\|_2^2$ arises from the closed-form rule $d_i = 1/\sigma_{t,i}^2$ not exactly recovering the marginal variances. By Lemma G.2, $\|\eta_t\|_2^2 \le N\bar{\eta}_t^2$. This term does *not* vanish with $\Delta t$ — it depends on the prior conditioning and accumulated data precision.

The diagonal mismatch is thus an $O(1)$-in-$\Delta t$ contribution to $\varepsilon_t$, but it is typically small: $\bar{\eta}_t = O(\kappa_s^2/d_{\min,t})$, and $d_{\min,t}$ stabilizes at a moderate value in steady state. For well-conditioned priors, $\|\eta_t\|_2^2$ is negligible compared to the off-diagonal term.

### G.5. Total Projection Error

**Lemma G.6** (Total covariance removal). *Assume $\|\widetilde{\Sigma}_t\|_{\mathrm{op}} \le C_\Sigma$ for all $t$ and $\Delta t$ sufficiently small. Then:*

$$\sum_{t=1}^{K} \mathrm{tr}(\Delta_t) \le C_\Sigma N + K|\tau|_{\max} + O(NT), \tag{51}$$

*where $|\tau|_{\max} := \max_t |\tau_t|$ with $\tau_t := \mathrm{tr}(\widetilde{\Sigma}_t) - \mathrm{tr}(\Sigma_t^+) = \sum_i \eta_{t,i}$ the trace discrepancy from the diagonal mismatch.*

*Proof.* From (45) and (46): $\Sigma_t^+ = A\widetilde{\Sigma}_{t-1}A^\top + \Sigma_\eta - \Delta_t$. Taking traces:

$$\mathrm{tr}(\Sigma_t^+) = \mathrm{tr}(A^\top A\,\widetilde{\Sigma}_{t-1}) + \mathrm{tr}(\Sigma_\eta) - \mathrm{tr}(\Delta_t).$$

For the IWP, $A = I + F\Delta t + O(\Delta t^2)$, so $\mathrm{tr}(A^\top A\,\widetilde{\Sigma}_{t-1}) = \mathrm{tr}(\widetilde{\Sigma}_{t-1}) + O(N\Delta t)$ (the perturbation $\Delta t\,\mathrm{tr}((F + F^\top)\widetilde{\Sigma}_{t-1})$ scales with the state dimension). Using $\mathrm{tr}(\widetilde{\Sigma}_t) = \mathrm{tr}(\Sigma_t^+) + \tau_t$:

$$\mathrm{tr}(\widetilde{\Sigma}_t) - \tau_t = \mathrm{tr}(\widetilde{\Sigma}_{t-1}) + \mathrm{tr}(\Sigma_\eta) - \mathrm{tr}(\Delta_t) + O(\Delta t). \tag{52}$$

Rearranging: $\mathrm{tr}(\Delta_t) = \mathrm{tr}(\widetilde{\Sigma}_{t-1}) - \mathrm{tr}(\widetilde{\Sigma}_t) + \tau_t + \mathrm{tr}(\Sigma_\eta) + O(\Delta t)$. Summing from $t = 1$ to $K$ and telescoping:

$$\sum_{t=1}^{K} \mathrm{tr}(\Delta_t) = \mathrm{tr}(\widetilde{\Sigma}_0) - \mathrm{tr}(\widetilde{\Sigma}_K) + \sum_{t=1}^{K} \tau_t + \sum_{t=1}^{K} [\mathrm{tr}(\Sigma_\eta) + O(N\Delta t)]$$

$$\leq C_\Sigma N + K|\tau|_{\max} + O(NT), \tag{53}$$

using $\mathrm{tr}(\widetilde{\Sigma}_0) \leq C_\Sigma N$, $\mathrm{tr}(\widetilde{\Sigma}_K) \geq 0$, and $\mathrm{tr}(\Sigma_\eta) = O(N\Delta t)$ for the IWP process-noise covariance. $\qquad\square$

**Proposition G.7** (Total projection error). *Under the assumptions of Lemma G.6, and writing $C' := (\beta + \delta_{\max})^2 C_Q^2/2$:*

$$\sum_{t=1}^{K} \varepsilon_t \leq C'\left(\max_t \|\Delta_t\|_{\mathrm{op}}\right)\sum_{t=1}^{K} \mathrm{tr}(\Delta_t) + \frac{(\beta + \delta_{\max})^2}{2}\sum_{t=1}^{K}\|\eta_t\|_2^2. \tag{54}$$

*Proof.* From Lemma G.4(c): $\varepsilon_t \leq C'\|\Delta_t\|_F^2 + \frac{(\beta+\delta_{\max})^2}{2}\|\eta_t\|_2^2$. For the first term: $\|\Delta_t\|_F^2 \leq \|\Delta_t\|_{\mathrm{op}}\,\mathrm{tr}(\Delta_t)$ since $\Delta_t \succeq 0$. Taking the max and summing gives the stated bound. For the second term: sum directly. $\qquad\square$

## G.6. Accumulated Error via Contraction

**Assumption G.8** (Filter stability). There exist $\rho \in [0, 1)$ and $C_0 > 0$ such that for any two Gaussian filtering distributions propagated through one prediction (45) and conditioning (46) step on the same FVM observation:

$$\mathrm{KL}(q_{t+1}\|p_{t+1}) \leq \rho \cdot \mathrm{KL}(q_t\|p_t) + C_0 \cdot \varepsilon_t. \tag{55}$$

*Remark* G.9 (When does this hold?). The contraction $\rho < 1$ follows from detectability of the pair $(A, H)$: unobserved modes must be dynamically stable. For the IWP+FVM system, the FVM constraint observes linear combinations involving the time-derivative components of the IWP state at each cell, and the IWP dynamics couples these to function values via $u_{t+1} \approx u_t + \Delta t\, u_t'$. Over $O(q)$ steps, this renders the full state observable. Detectability holds under the same conditions needed for the PDE solver to be meaningful (well-posed PDE, adequate spatial resolution).

**Proposition G.10** (Accumulated error bound). *Let $q_t$ denote the exact filtering distribution and $p_t$ the moment-matched approximate distribution. Under Assumption G.8 with $q_0 = p_0$:*

(i) *Pointwise bound.* At any time step $t$:

$$\mathrm{KL}(q_t\|p_t) \leq \frac{C_0}{1-\rho} \max_{s \leq t} \varepsilon_s. \tag{56}$$

(ii) *Summation bound.* For any $t$:

$$\mathrm{KL}(q_t\|p_t) \leq C_0 \sum_{s=0}^{t-1} \rho^{t-1-s}\, \varepsilon_s. \tag{57}$$

(iii) *Riccati steady state.* In steady state, $\Delta_t \to \Delta_\infty$ with $\|\Delta_\infty\|_{\mathrm{op}} = O(\Delta t)$, and $\eta_t \to \eta_\infty$ with $\|\eta_\infty\|_2$ independent of $\Delta t$. The accumulated error converges to:

$$\lim_{t\to\infty} \mathrm{KL}(q_t\|p_t) \leq \frac{C_0}{1-\rho}\left(O(\Delta t^2) + \varepsilon_{\mathrm{diag}}\right), \tag{58}$$

*where $\varepsilon_{\mathrm{diag}} := \frac{(\beta+\delta_{\max})^2}{2}\|\eta_\infty\|_2^2$ is independent of $\Delta t$.*

*Proof.* **Part (i).** Unroll (55) from $\mathrm{KL}(q_0\|p_0) = 0$:

$$\mathrm{KL}(q_t\|p_t) \le C_0 \sum_{s=0}^{t-1} \rho^{t-1-s}\varepsilon_s \le \frac{C_0}{1-\rho}\max_s \varepsilon_s.$$

**Part (ii)** is the intermediate expression.

**Part (iii).** In steady state, $\Sigma_t^- \to \Sigma_\infty^-$, $\Sigma_t^+ \to \Sigma_\infty^+$. From the prediction equation with $A = I + F\Delta t + O(\Delta t^2)$:

$$\Sigma_\infty^- = A\widetilde{\Sigma}_\infty A^\top + \Sigma_\eta = \widetilde{\Sigma}_\infty + \Delta t(F\widetilde{\Sigma}_\infty + \widetilde{\Sigma}_\infty F^\top) + O(\Delta t^2).$$

Standard Riccati-equation analysis for the analogous linear-Gaussian filter shows that under bounded gain $G_\infty$ and Assumption G.8, the steady-state update covariance satisfies $\|\Delta_\infty\|_{\mathrm{op}} = O(\Delta t)$; we do not reproduce this argument here. Since $\Delta_\infty = G_\infty S_\infty^{-1} G_\infty^\top$ has rank at most $m$ (the observation dimension per step), this gives $\|\Delta_\infty\|_{\mathrm{F}}^2 = O(m\,\Delta t^2)$. The diagonal mismatch $\eta_\infty$ depends on $Q_s$ and $d_{\min,\infty}$, but *not* on $\Delta t$. Taking $t \to \infty$ in (56) gives (58) with the $O(\Delta t^2)$ term absorbing the dependence on $m$ and the constants from Lemma G.4(c). □

*Remark* G.11 (Interpretation). The accumulated error decomposes into an $O(\Delta t^2)$ component from the off-diagonal projection error (which vanishes with temporal refinement) and a floor $\varepsilon_{\mathrm{diag}}$ from the diagonal mismatch (which persists even as $\Delta t \to 0$). The floor is controlled by $\kappa_s$ and $d_{\min,\infty}$, and inherits the dimension dependence from $\|\eta_\infty\|_2^2 \le N\bar{\eta}_\infty^2$ in the worst case; in practice, when the per-coordinate mismatches $\eta_{\infty,i}$ remain bounded as the mesh is refined, $\varepsilon_{\mathrm{diag}}$ stays small compared to the $O(\Delta t^2)$ term for any reasonable $\Delta t$, which is consistent with the empirical accuracy observed in our experiments.

