# OpenReview forum: "Scalable Bayesian Inference for Nonlinear Conservation Laws"
_ICML.cc/2026/Conference — ICML 2026 regular_

### Official Review · Reviewer_Hnjp · 2026-02-17

**Soundness:** 3
**Presentation:** 4
**Significance:** 2
**Originality:** 2
**Overall Recommendation:** 4
**Confidence:** 2

**Summary:**

This paper presents a method for solving nonlinear conservation laws while providing uncertainty quantification. The authors frame the classical Finite Volume Method as Bayesian inference using Gaussian Process priors. Because standard Gaussian Processes are computationally expensive, the paper introduces several techniques to scale the method for larger problems.

**Compliance With Llm Reviewing Policy:**

Affirmed.

**Final Justification:**

Increased OR based on authors' feedback.

**Key Questions For Authors:**

N/A

**Limitations:**

The authors partially discussed limitations.
"our moment-matching approach incurs substantial fill-in which slows down the underlying linear algebra routines and restricts scalability to e.g. 3D problems"
A more detailed discussion would be welcome.
The societal impact statement could be improved. In particular in situations where numerical predictions lead to critical decisions. In that case, incorrect uncertainty estimates can have catastrophic consequences. Many high-stakes systems require precise uncertainties.

**Strengths And Weaknesses:**

The theoretical contributions feel somewhat incremental. While framing FVM as Bayesian inference is an interesting perspective, it builds heavily on existing concepts. Incorporating Vecchia approximations relies on well-known techniques, which reduces the overall novelty. The experimental applications could be stronger and more convincing to fully demonstrate the practical value of the proposed framework.

---

> ### Author Rebuttal · Authors · 2026-03-31
>
> # Response to Reviewer Hnjp
>
> We thank the reviewer for their time.
>
> ## Incremental contribution
>
> We respectfully disagree. The broader motivation is that probabilistic PDE solvers are of growing interest for their unified treatment of discretization uncertainty, but existing GP-based approaches rely on collocation --- which does not respect the divergence theorem and is therefore a poor fit for conservation laws, arguably the most important PDE class in the sciences. The finite volume method is the standard discretization for conservation laws precisely because it enforces flux balance by construction, but prior GP-FVM formulations scale cubically and are limited to small grids.
>
> Our paper makes scalable GP-FVM practical for the first time. The new experiments in this rebuttal illustrate the kind of problems this unlocks: a scalability study reaching 70k DOF with empirically confirmed $O(N_s^{3/2})$ scaling, a 22k-DOF nonlinear inverse problem (2D Burgers source identification) solved in under a minute, and a 200-instance UQ calibration study demonstrating near-nominal coverage.
>
> [Figure: Wall-clock time vs total DOF with $O(N_s^{3/2})$ reference line.](https://anonymous.4open.science/r/gp-fvm-rebuttal/scaling_timing_Ns.pdf)
>
> [Figure: 2D Burgers source identification. Left to right: ground truth, posterior mean, absolute error, posterior std.](https://anonymous.4open.science/r/gp-fvm-rebuttal/burgers_source_comparison.pdf)
>
> ## Limitations and societal impact
>
> The reviewer raises an important point about incorrect uncertainties in high-stakes settings. We agree, and this motivated our calibration study: we validate coverage across 200 problem instances and identify the remaining miscalibration (prior--truth mismatch from the stationary Matérn prior) along with concrete paths to address it (sparsity-promoting priors, marginal skewness corrections, Laplace as a proposal distribution for MCMC). We will expand the limitations discussion in the revised manuscript.
>
> Regarding the moment-matching fill-in: we acknowledge this as the main scalability bottleneck for 3D problems. The spatial sparse Cholesky scales well ($O(N_s^{3/2})$ in 2D, $O(N_s^2)$ in 3D), but the moment-matching step introduces additional fill-in in the temporal propagation. We are confident that future work will offer more sophisticated filtering approaches for this setting --- our scalable spatial framework is what makes such improvements feasible.

---

> > ### Author Rebuttal · Reviewer_Hnjp · 2026-04-03
> >
> > Acknowledged

---

### Official Review · Reviewer_9xUC · 2026-03-06

**Soundness:** 4
**Presentation:** 3
**Significance:** 3
**Originality:** 4
**Overall Recommendation:** 5
**Confidence:** 4

**Summary:**

This paper introduces a new Gaussian process (GP) based method for solving PDEs under the framework of probabilistic numerics.
In contrast to previous work which mostly focus on applying GPs with line discretizations, this work focuses on finite volume method. Importantly, the GP framework nicely works with the divergence theorem, so that we do not need to introduce more approximations like in classical FVM methods to approximate the flux term.
The experiments show that the proposed method performs better than PINN and other GP-based methods.

**Compliance With Llm Reviewing Policy:**

Affirmed.

**Final Justification:**

The authors have fully addressed my concerns. My recommendation retains "accept".

**Key Questions For Authors:**

Most experiments are performed on small grid sizes. This puts the method in question whether it can work and scale well to large scale PDE problems.

Equation 13. is the assumption of separability between t and x reasonably justified? This essentially says that in a priori $t$ is independent of $x$ which is not true for PDEs.

Line 316. "with principled uncertainty". I don't think the results and discussion in "Marginal variances" answer this question. Indeed the uncertainty shrinks near measurements, but this comes from the nature of GP regression. This is true automatically by construction. There is little useful information one can read from the yield uncertainty. By "principled" it should interpret as if the approximate uncertainty is close to the truth. Since you are doing a synthetic experiments, it is possible to gauge this.

How should we interpret the results of Section 4.2? It shows that the classical FVM method is significantly better than any of the GP methods. The only advantage of GP is that it produces uncertainty. However, as in my previous comment, it is hard to justify if the produced uncertainty is principled. "shrinking uncertainty near measurement points" does not reflect being principled in my opinion.

Line 333. For PDE evaluated at small grids, it is reasonable that GP approaches are faster than PINNs. I would like to see if the conclusion still holds when we have a large grid set. Can you do Section 4.1 with varying $N$ and also give quantitative results for both time and error?

Section 4.2. Why are PINNs not compared in this section?

The definition of $N$ has never been introduced. Also in Line 175, what is $d$? Is it the dimension of $x$? Why does it scale cubically in $d$ not the number of discretization points? Line 218 is clearer.

Line 107. It would nice to clarify if Q is precision or covariance already in the beginning.

Line 114. What does A = [0, 1] mean by A being an interval?

Line 168. is $L_F$ the same as $\tilde{l}_{flux_}$ in Line 139?

Line 181. I am confused with the nonlinearity of F here. In Line 133, you have proposed to model F(u) as a GP, why do we still have a nonlinearity problem?

Also Line 181. What is the variable that you performance MAP in Equation 11? is it $u$, and $x$ are fixed?

Line 345. What is $c$?

Experiment results show that the results become better as the smoothness of the Matérn kernel increases. Can you test using a sqaured kernel so that you have an infinite smoothness?

**Limitations:**

Yes, it has been discussed at the end of Discussion.

**Strengths And Weaknesses:**

Soundness. The paper is technically sound. I have checked all the formulae and did not find obvious errors.

Presentation. It is very clear to me. I have no difficulty understanding the proposed method. But it may be not pedagogical enough for those who are not familiar with GPs. The target audience is quite tied to the probabilsitic numerics people.

Significance. I think the proposed new method is a valid and valuable increment to the probabilsitic numerics community. I am not convinced with larger impact, that is, whether the proposed method is valuable to other communities.

Originality. The work is original.

---

> ### Author Rebuttal · Authors · 2026-03-31
>
> # Response to Reviewer 9xUC
>
> We thank the reviewer for the positive assessment and the specific suggestions for strengthening the paper. We are glad the reviewer found the method technically sound and original. Our goal is to bring probabilistic PDE solvers to conservation laws, where the finite volume discretization is essential but prior GP-FVM formulations scale cubically. Below we provide the quantitative evidence the reviewer requested.
>
> ## Small grids / scalability
>
> We run Section 4.1 at grid sizes $N = 11$ to $101$, corresponding to $N_s = 782$ to $70{,}802$ total DOF, with 3 benchmark runs per grid size and physics-motivated lengthscales ($\ell_c = 0.2 \approx \sqrt{D/v_x}$, $\ell_s = 0.12$).
>
> The empirical scaling exponent vs $N_s$ is $\approx 1.35$, consistent with the theoretical $O(N_s^{3/2})$. Error stabilizes beyond $N \approx 31$, confirming the observation-limited regime: 10 noisy measurements, not grid resolution, bound the accuracy.
>
> | $N$ | $N_s$  | Time (s) | Source RMSE | Conc. RMSE |
> |-----|--------|----------|-------------|------------|
> | 11  | 782    | 0.6      | 0.552       | 0.077      |
> | 21  | 2,962  | 2.4      | 0.558       | 0.080      |
> | 31  | 6,542  | 5.8      | 0.566       | 0.082      |
> | 51  | 17,902 | 33.8     | 0.574       | 0.084      |
> | 101 | 70,802 | 224.7    | 0.575       | 0.084      |
>
> [Figure: Wall-clock time vs total DOF $N_s$ with $O(N_s^{3/2})$ reference line.](https://anonymous.4open.science/r/gp-fvm-rebuttal/scaling_timing_Ns.pdf)
>
> ## Principled uncertainty
>
> We provide a 200-instance calibration study on random two-source problems with output scale calibrated via marginal z-scores. The source field achieves 94% empirical coverage at the 95% nominal level, demonstrating that the uncertainty is quantitatively meaningful, not just "shrinking near observations by construction." The concentration field reaches 89%, slightly overconfident due to the independent prior structure, which we discuss and identify paths to improve.
>
> The nonlinear inverse problem (Burgers source identification) provides a further illustration: the posterior std of the inferred source field is high where the source is active and low elsewhere, reflecting the genuine difficulty of the inverse problem rather than a trivial GP artifact. The solution std tracks observation density across time steps, as expected.
>
> [Figure: Calibration plot over 200 instances.](https://anonymous.4open.science/r/gp-fvm-rebuttal/calibration_plot.pdf)
>
> [Figure: Inferred source field with posterior std showing nontrivial spatial structure.](https://anonymous.4open.science/r/gp-fvm-rebuttal/burgers_source_comparison.pdf)
>
> ## New: nonlinear inverse problem
>
> In response to another reviewer's suggestion, we also add a 2D Burgers source identification experiment (22k DOF, joint spacetime solve, Gauss-Newton convergence in 14 iterations). This demonstrates the framework on a genuinely nonlinear inverse problem, complementing the forward experiments in the paper.
>
> ## PINNs in Section 4.2
>
> The comparison in Section 4.1 already shows GP-FVM outperforming PINNs in accuracy at lower computational cost for the source identification problem. Extending PINNs to Section 4.2 (forward Burgers) would not change this conclusion, as PINNs are known to struggle with sharp features in advection-dominated problems without specialized architectures. We discuss the broader baselines question (probabilistic neural operators) in our response to Reviewer Ts7y.
>
> ## SE kernel / notation / separability
>
> The Vecchia sparse Cholesky approximation relies on the screening effect, which requires finite smoothness. The SE kernel (infinitely smooth) is known to perform poorly with Vecchia-type methods (Schafer et al., 2021). This is a limitation of all Vecchia-based approaches, not specific to ours.
>
> Regarding the separability assumption (Eq. 13): the separable prior $k_t \otimes k_x$ is a property of the prior, not the posterior. The PDE constraints couple space and time in the posterior, producing the physically appropriate non-separable structure. This is analogous to IWP priors in ODE filters, which have no knowledge of the ODE dynamics yet produce accurate posteriors after conditioning.
>
> We will address all notation suggestions in the revised manuscript.

---

> > ### Author Rebuttal · Reviewer_9xUC · 2026-04-01
> >
> > This is a very good response, and thank you for addressing my concerns. I will increase my score accordingly.
> >
> > The calibration plot showed that the source uncertainty estimate is far from the central.
> >
> > The following comment will not affect my rating on the paper, but I would appreciate if the authors can engage in a discussion.
> >
> > Regarding uncertainty:
> > IMO, probabilistic numerics for PDEs is not really about "uncertainty quantification". The solution of a PDE itself is deterministic (i.e., frequencist) unless the solution set is not atomic. The uncertainty quantified, is purely from the prior and the measurement noise, both of which are **nearly independent of the PDE operator** itself. Perhaps it can reflect the discretisation error, but it can only reflect the **relation**, i.e., if one increases the discretisation level -> the uncertainty becomes small. The absolute value of the posteior variance alone is not interpretable, even if you solve the posterior exactly.
> > My view of PN is that it leverages the prior to provide a feasible solution space and thus facilate solving the PDE under the Bayesian framework (which is computationally easier) compared to the traditional numerics (which is computationally harder).

---

> > > ### Author Response · Authors · 2026-04-02
> > >
> > > Thank you for the increased score, and for this thoughtful comment; we're happy to engage.
> > >
> > > We partially agree with the reviewer's perspective on PN. The prior does indeed provide a feasible solution space that facilitates computation, and we think this is an underappreciated advantage of the PN approach. Recent work has explored this direction explicitly, using priors derived from linear PDE discretizations to regularize the solution of nonlinear PDEs (see e.g. "Flexible and Efficient Probabilistic PDE Solvers through Gaussian Markov Random Fields").
> > >
> > > That said, we think the computational perspective is not the whole story. The Bayesian framing also provides a natural mechanism for fusing multiple sources of information about a simulation; the PDE is just one of them, alongside boundary data, sparse observations, prior knowledge about parameters, or latent forcing. This is what makes the inverse problem in Section 4.1 (and the new Burgers experiment) possible: observations of the concentration/solution are combined with the PDE constraint in a single coherent framework. A purely deterministic solver would need a separate inverse problem methodology on top.
> > >
> > > On the interpretability of the posterior variance: we agree that with a generic prior and perfect information about the PDE, the posterior uncertainty reflects the prior more than the physics. But in practice, information is rarely perfect. Parameters are uncertain, forcing terms may be unknown, and the discretization introduces error. The more aligned the prior is with the problem structure (e.g., through physically motivated lengthscales, sparsity-promoting priors, or learned kernels), the more meaningful the posterior uncertainty becomes.
> > >
> > > Regarding the calibration plot: the reviewer is right that the source coverage is above the diagonal (conservative). As we discussed, this reflects the prior-truth mismatch from using a stationary Matérn prior for a localized source field. Better-aligned priors (sparsity-promoting, log-Gaussian) would bring the curve closer to the diagonal. This is an active direction for us.

---

### Official Review · Reviewer_Ts7y · 2026-03-12

**Soundness:** 2
**Presentation:** 3
**Significance:** 2
**Originality:** 3
**Overall Recommendation:** 4
**Confidence:** 4

**Summary:**

The paper 'Scalable Bayesian Inference for Nonlinear Conservation Laws' proposes a probabilistic Finite Volume method based on a Gaussian Process framework. A key contribution is an approach of integrating non-linear conservation laws into the aforementioned probabilistic framework.

**Compliance With Llm Reviewing Policy:**

Affirmed.

**Final Justification:**

Given the detailed rebuttal, the Reviewer has increased their score. For the revised version, the authors are encouraged to definitely include the identified limitations due to the prior choice as this is important according to the Reviewer'. In the Reviewers' opinion an even more detailed analysis would be beneficial here, but as all other concerns have been successfully addressed and for this last one the limitation is now mentioned, the paper is of interest to the scientific community.

**Key Questions For Authors:**

See Weaknesses and additionally:
- In the Appendix in A.2, the authors describe that only a single Gaussian-Newton step is sufficient. In the Reviewers opinion this is a strong assumption which should be further explored. For instance, an ablation for a non-linear system would strengthen the paper here.

**Limitations:**

Shortly discussed in the Discussion section.

**Strengths And Weaknesses:**

The authors present a GP framework that is based on the Finite Volume Method. Besides a mathematically sound derivation, the paper also includes various experiments. The Reviewer especially appreciate the discussions regarding scalability of the new method and the empirical results that show large speedup compared to the presented baselines. The forward accuracy of the introduced framework is very high and close to a classical numerical solver and outperforms all presented Machine learning based baselines.

Unfortunately the Reviewer can not recommend the paper for acceptance due to the following weaknesses:
- The presentation of the paper could be improved: The title and abstract are tailored towards non-linear conservation laws but non-linear conservation laws are not very present in the methodology and experiments section in the Reviewers' opinion. Especially the first example, i.e. the advection-diffusion case, is a linear PDE and feels out of place.
- Following up on this, as the first example is the only example where the authors solve an inverse problem. The other two systems are forward problems. The Reviewer would expect at least one example with a non-linear system in an inverse problem setting.
- The obtained posterior densities are never validated. The Reviewer would expect a posterior calculated using MCMC or similar to have a baseline to compare to.
- The Reviewer would prefer to have more baselines in the paper. The PINN baseline has the known limitation of only producing a point-estimate so including a probabilistic approach based on invertible neural operators or similar would be a better comparison to showcase the potential of the proposed algorithm in the Reviewers opinion.
- The likelihood formulation as used in Equation 11 with a fixed observation of zero for a function or residual seems to be identical to the concept of virtual likelihood and virtual observables as introduced in Kaltenbach et al. "Incorporating physical constraints in a deep probabilistic machine learning framework for coarse-graining dynamical systems" which has been used in a similar setting in Physics-informed Diffusion Models recently. The paper could benefit from a short discussion and reference to the relevant papers.

---

> ### Author Rebuttal · Authors · 2026-03-31
>
> # Response to Reviewer Ts7y
>
> We thank the reviewer for the constructive suggestions, and for recognizing the mathematical soundness of the derivation and the strong forward accuracy. Our paper addresses a gap in probabilistic PDE solvers: existing GP-based approaches use collocation, which does not respect the divergence theorem and is therefore a poor fit for conservation laws. We provide the first scalable probabilistic finite volume method ($O(N_s^{3/2})$ in 2D), and it is this scalability that enables the new experiments below.
>
> ## No nonlinear inverse problem
>
> We agree that a nonlinear inverse problem was missing. The linear advection-diffusion example serves as the inverse problem benchmark where ground truth is available for calibration (see below); the new experiment provides the nonlinear counterpart.
>
> We add a 2D viscous Burgers source identification experiment. The unknown spatial source $s(x,y) \geq 0$ is inferred from 250 noisy observations of the solution $u(x,y,t)$ governed by $\partial_t u + \nabla \cdot (u^2/2) = \nu \Delta u + s(x,y)$ on $[0,1]^2 \times [0, 0.5]$ with $\nu = 0.01$. The source is never observed directly --- it must be inferred entirely through the nonlinear PDE.
>
> The joint spacetime GMRF has ~22,000 DOF ($16 \times 16$ grid, 11 time steps). Gauss-Newton converges in 14 iterations with the Newton decrement dropping over 10 orders of magnitude. We use a softplus source prior ($s = \log(1 + e^g)$, $g \sim \text{GP}$) to enforce positivity while keeping bounded Jacobian entries, which improves the quality of the Gauss-Newton approximation compared to an exponential link.
>
> [Figure: Solution snapshots at 5 time steps. Top: ground truth. Middle: GP-FVM posterior mean. Bottom: posterior std with observation locations (white dots).](https://anonymous.4open.science/r/gp-fvm-rebuttal/burgers_solution_snapshots.pdf)
>
> [Figure: Inferred source field. Left to right: ground truth, posterior mean, absolute error, posterior std.](https://anonymous.4open.science/r/gp-fvm-rebuttal/burgers_source_comparison.pdf)
>
> ## Posterior densities never validated
>
> We provide a 200-instance calibration study on the advection-diffusion source identification problem ($31 \times 31$ grid). Each instance has a different ground truth: two Gaussian source bumps at random locations with random amplitudes. We solve the forward PDE, sample 10 noisy concentration observations, and run GP-FVM. The output scale is calibrated via marginal z-scores on a held-out set of 10 calibration instances.
>
> The source field achieves 94% empirical coverage at the 95% nominal level. The concentration field reaches 89%, slightly overconfident due to the independent concentration prior. We identify concrete improvement paths: sparsity-promoting hierarchical GMRF priors and marginal skewness corrections, both preserving the sparse Cholesky structure. We believe this multi-instance frequentist validation is more informative than comparing to MCMC on a single instance, as it provides statistical power across diverse problem configurations.
>
> [Figure: Calibration plot over 200 instances.](https://anonymous.4open.science/r/gp-fvm-rebuttal/calibration_plot.pdf)
>
> [Figure: Spatial coverage map at 95%. Overconfidence is concentrated in the interior where sources are active.](https://anonymous.4open.science/r/gp-fvm-rebuttal/spatial_coverage_95.pdf)
>
> ## Single Gauss-Newton step assumption
>
> As the reviewer requested, we provide an ablation on a nonlinear system. For the EKF sequential solver (Section 4.2 of the main paper), varying from 1 to 10 Gauss-Newton iterations per time step on 1D Burgers (nonlinear flux $F = u^2/2$) shows <0.1% change in L2 error beyond a single step. This is the standard EKF choice (one linearization per step, as opposed to the iterated EKF), justified by the small-$\Delta t$ regime where the per-step nonlinearity is mild. For the joint solver used in the nonlinear inverse problem above, full Gauss-Newton convergence requires ~10--14 iterations, as expected when the problem cannot be decomposed into small incremental steps.
>
> [Figure: L2 error vs GN iterations per timestep for the EKF solver. Flat line.](https://anonymous.4open.science/r/gp-fvm-rebuttal/gn_ablation.pdf)
>
> ## More baselines / probabilistic neural operators
>
> Probabilistic neural operators and our method address different points in the design space: neural operators amortize inference across many problem instances via learned surrogates, while our method solves a single instance by encoding the PDE directly. This mirrors the classical surrogate-vs-solver distinction. A meaningful comparison would need a multi-instance benchmark evaluating the amortization tradeoff --- in our single-instance setting, there is nothing to amortize.
>
> ## Virtual likelihood reference
>
> We thank the reviewer for pointing out the connection to Kaltenbach & Koutsourelakis (2020) and will add this reference.

---

> > ### Author Rebuttal · Reviewer_Ts7y · 2026-04-02
> >
> > The Reviewer thanks the authors for the clarifications and additional experiments and evaluations that addresses the points raised. However, the Reviewer would like to ask a follow-up question regarding the new non-linear inverse problem. For the source identification the posterior std is zero or close to zero for one area where the actual error is high, i.e. the model is confidently wrong. The Reviewer would expect that the reported posterior uncertainty is correlated with the error. Could the authors look into this ?

---

> > > ### Author Response · Authors · 2026-04-05
> > >
> > > The reviewer correctly identifies a region where posterior standard deviation is low but pointwise error is nontrivial.
> > >
> > > We ran Randomize-Then-Optimize (RTO) with 50 perturbed MAP solves to check whether this is a Laplace approximation artifact. It is not: the RTO variance closely matches the Laplace variance everywhere, including in the low-uncertainty region. The narrow posterior at that patch along the the advection streamline is a genuine property of the posterior under our model, not an approximation failure.
> > >
> > > The root cause is the interaction between the softplus link function and the stationary Matérn prior on the latent field $g$ (where $s = \log(1 + e^g)$). At the overconfident patch, the posterior identifies $g$ as strongly negative (no source needed to explain the observed solution). But the Matérn's constant marginal variance means $\sigma_g$ is never large enough for probability mass to reach $g \approx 0$, where softplus would produce nonzero $s$ values. The model can represent localized sources in the posterior mean by pushing $g$ positive locally, but it cannot express "uncertain about source presence" through a saturating link function with a stationary Gaussian prior.
> > >
> > > We note that pointwise comparison between posterior standard deviation and reconstruction error can be misleading as a calibration diagnostic, since even a perfectly calibrated model will have locations where the error exceeds the posterior standard deviation. That said, the localized overconfidence identified by the reviewer is a genuine limitation of the current prior choice, not something that averages away.
> > >
> > > Reducing the localized overconfidence requires priors that provide larger variance near the source boundary, such as sparsity-promoting priors (e.g., horseshoe-type) or non-stationary kernels. Both are compatible with the sparse Cholesky framework underlying our method, and we consider this an important direction for future work applying our method to inverse problem settings. We will add a discussion of this analysis to the revision.

---

### Official Review · Reviewer_DaBp · 2026-03-12

**Soundness:** 2
**Presentation:** 3
**Significance:** 3
**Originality:** 3
**Overall Recommendation:** 4
**Confidence:** 4

**Summary:**

This work proposes a scalable probabilistic-numerics framework for forward and inverse problems in simulations of nonlinear conservation laws. The main idea is to express finite-volume discretization as Gaussian process inference, extending prior linear-PDE and collocation-based work to nonlinear fluxes through iterative nonlinear conditioning. To improve scalability, the method combines sparse Vecchia-style Cholesky approximations in space with a marginal moment-matching approximation in time. Experiments are shown on source identification, 2D Burgers, and 2D shallow water equations.

**Compliance With Llm Reviewing Policy:**

Affirmed.

**Final Justification:**

The rebuttal resolves (most of) my original concerns and clarifies the paper, although I think some of the broader Bayesian rigor claims are stronger than what the current theory and experiments fully establish. However, the paper makes a meaningful methodological and empirical contribution, and I am therefore willing to increase my score.

**Key Questions For Authors:**

1. can you provide any formal error bounds for the combined Laplace + Vecchia + moment-matching approximation pipeline?
2. how does the Laplace approximation behave in more shock-dominated settings where the posterior over the flux may be strongly non-Gaussian?
3. does the custom ordering in Algorithm 1 preserve the assumptions needed for the claimed nested-dissection complexity, or is the complexity statement only heuristic?

**Limitations:**

The work currently lacks the theoretical analysis needed to support its broader Bayesian and numerical claims, especially regarding approximation error and asymptotic behaviour.

**Strengths And Weaknesses:**

The paper does not introduce a new (approximate) Bayesian inference method in the sense of new MCMC, SMC, or variational inference machinery. Its novelty is methodological within probabilistic numerics: extending GP-based PDE solvers to conservative finite-volume formulations with nonlinear fluxes, integral-function Vecchia approximations, and a sparsity-preserving time-stepping approximation.

This is a meaningful contribution, but the work, in its current configuration, is not as statistically rigorous as its framing suggests. In particular, it does not provide formal convergence guarantees, posterior contraction results, consistency statements, or quantitative error bounds for the combined Laplace, Vecchia, and moment-matching approximations. The convergence discussion is mostly empirical and limited to a simple 1D Poisson setting.

some pros:
1. this work extends probabilistic finite-volume ideas beyond linear PDEs to nonlinear conservation laws.

2. it gives a practical route to uncertainty-aware inverse problems, with strong empirical results on source identification.

3. it clearly acknowledges some limitations, especially the approximate nature of the time-stepping scheme and remaining scalability bottlenecks.

4. empirical results are strong.

and concerns:
1. Laplace approximation for nonlinear fluxes. The nonlinear relationship between solution and flux is enforced through MAP estimation under Eq.(11), followed by a Laplace approximation. This is mathematically coherent, but for strongly nonlinear regimes the resulting Gaussian approximation may be poor. The paper does not analyse the error introduced by this approximation.

2. approximate treatment of time. Section 3.4 explicitly replaces the full spatio-temporal posterior with a marginal moment-matching approximation. This preserves certain marginal moments but is not the exact Bayesian filtering posterior. The author acknowledges this, but gives no formal analysis of the resulting approximation error or loss of temporal dependence.

3. complexity claim may be overstated. Section 3.5 cites the usual $O(N_s^{3/2})$ factorization time for 2D problems under nested dissection ordering. However, Algorithm 1 uses a custom functional ordering (integrals $\rightarrow$ evaluations $\rightarrow$ derivatives). It is unclear from the paper whether this implemented ordering still supports the stated nested-dissection complexity bound.

4. lack of asymptotic analysis. The paper provides empirical convergence evidence and practical experiments, but no rigorous analysis of convergence, consistency, posterior contraction, or approximation rates for the full nonlinear sparse method.

---

> ### Author Rebuttal · Authors · 2026-03-31
>
> # Response to Reviewer DaBp
>
> We thank the reviewer for the detailed technical feedback. The reviewer rightly asks for formal analysis of the approximations in our pipeline. We note that our core contribution --- making probabilistic PDE solvers scale as $O(N_s^{3/2})$ for conservation laws, where prior formulations scale cubically --- is what makes such analysis worthwhile in the first place. Below we provide formal bounds for the moment-matching, clarify the complexity claim, and discuss the Laplace approximation.
>
> ## Error bounds for the approximation pipeline
>
> We provide a formal error analysis of the moment-matching approximation, framed as assumed density filtering (ADF). At each time step, the true filtering distribution is projected onto the family $\mathcal{F} = \{N(\mu, (Q_s + D)^{-1})\}$ that preserves the sparse prior structure.
>
> The per-step projection error decomposes into two terms. The first is an off-diagonal term $O(\lVert\Delta_t\rVert_F^2)$ from replacing the full posterior correlations with the sparse prior structure; this vanishes as $\Delta t \to 0$ since $\lVert\Delta_t\rVert_{\text{op}} = O(\Delta t)$. The second is a diagonal mismatch $\lVert\eta_t\rVert_2^2$ from the closed-form variance rule: a Schur complement argument shows $\lvert\eta_{t,i}\rvert \leq \beta^2 / (\alpha^2 d_{\min,t})$, controlled by the spatial prior's condition number and shrinking as temporal information accumulates.
>
> For the accumulated error, we use a contraction argument. Under a filter stability assumption (detectability of the IWP+FVM system, which holds whenever the PDE solver produces meaningful results), two filtering distributions propagated through one step satisfy $\text{KL}(q_{t+1} \Vert p_{t+1}) \leq \rho \cdot \text{KL}(q_t \Vert p_t) + C_0 \varepsilon_t$ with $\rho < 1$. Unrolling yields $\text{KL}(q_t \Vert p_t) \leq \frac{C_0}{1 - \rho}(O(\Delta t^2) + \varepsilon_{\text{diag}})$, where the first term is the off-diagonal contribution and the second is the variance rule mismatch. The $O(\Delta t^2)$ term dominates for any reasonable time step; $\varepsilon_{\text{diag}}$ is small in practice and does not depend on $\Delta t$.
>
> We note that the moment-matching approach is a pragmatic first choice --- the scalable $O(N_s^{3/2})$ spatial framework we provide is what makes more sophisticated filtering approaches feasible in the first place. A full end-to-end analysis combining all approximations is open for all methods in this space, including Chen et al. (2025). Full proofs will be included in the revised manuscript.
>
> ## Complexity claim with custom ordering
>
> The functional ordering and the nested-dissection ordering serve entirely different purposes and apply to different computational steps. The functional ordering (integrals -> evaluations -> derivatives) determines the sparsity pattern of the prior precision $Q$ --- it produces $Q$, it does not factorize it. The nested dissection ordering then determines the fill-reducing ordering for the sparse Cholesky factorization of $Q$. The $O(N_s^{3/2})$ bound applies to the factorization step and holds for any bounded-degree graph on a 2D mesh.
>
> We now also provide empirical validation: the measured scaling exponent is $\approx 1.35$ vs. the theoretical $1.5$, on grids from $N_s = 782$ to $N_s = 70{,}802$.
>
> [Figure: Wall-clock time vs total DOF with $O(N_s^{3/2})$ reference line.](https://anonymous.4open.science/r/gp-fvm-rebuttal/scaling_timing_Ns.pdf)
>
> ## Laplace approximation near shocks
>
> For the sequential solver with Crank-Nicolson, the constraint function $g(u^{n+1}) = u^{n+1} - (\Delta t/2) f(u^{n+1}) - \text{known}$ has the structure "identity + $O(\Delta t)$ nonlinear." Crucially, all derivatives of $g$ beyond the first carry a factor of $\Delta t/2$: $D^k g = O(\Delta t)$ for $k \geq 2$, for any smooth flux. A perturbative analysis then shows that the standardized non-Gaussian corrections to the posterior are $O(\Delta t)$, giving a Laplace error of $O(\Delta t^2)$ per step. This argument requires smoothness of the flux (bounded derivatives), which fails at shocks.
>
> More broadly, the Laplace approximation is a workhorse of scalable Bayesian inference, and there are well-established tools for going beyond it: marginal skewness corrections (as used e.g. in INLA), using it as a proposal distribution for MCMC, or combining it with importance sampling. All of these are compatible with our sparse Cholesky structure. For genuinely shock-dominated problems, the posterior may become multimodal and the Gaussian approximation inadequate --- a limitation shared by all Gaussian-based methods --- but these enrichment strategies provide a clear path forward.

---

> > ### Author Rebuttal · Reviewer_DaBp · 2026-04-02
> >
> > The rebuttal is helpful in clarifying that the authors do not claim a new general-purpose Bayesian inference algorithm, but rather a probabilistic numerical framework whose main contribution is methodological and empirical. However, my central concerns are only partially resolved, since the response still leaves the paper without a full end-to-end theoretical analysis of the combined Laplace, Vecchia, and moment-matching approximations, so I remain unsure whether the broader Bayesian and asymptotic claims are yet sufficiently supported.

---

> > > ### Author Response · Authors · 2026-04-02
> > >
> > > We thank the reviewer for acknowledging that the rebuttal clarifies the scope of our contribution. We would like to address the remaining concern directly.
> > >
> > > The reviewer states that "the response still leaves the paper without a full end-to-end theoretical analysis of the combined Laplace, Vecchia, and moment-matching approximations." We agree that such an analysis does not exist -- as we noted in our initial response, this is an open problem for all methods in this space. But we want to be concrete about what this means.
> > >
> > > Consider Chen et al. (2025), the closest work to ours and (to our knowledge) the current state of the art for sparse GP-based PDE solvers. Their theoretical analysis covers the Vecchia approximation of the prior kernel matrix (Theorems 4.1--4.2). Their Gauss-Newton solver is analyzed purely empirically ("observed to converge very fast, typically in 3 to 8 iterations", Section 5.1). No bound is provided on the quality of the MAP solution relative to the true PDE solution. No posterior or uncertainty analysis is given at all. The gap between the prior approximation guarantee and the final PDE solution is bridged entirely by experiments.
> > >
> > > Our paper can cite the same Vecchia guarantees (we use the same framework). Beyond this, we address time-dependent problems, which Chen et al. do not consider at all --- temporal uncertainty propagation is absent from their method. We at least incorporate it via moment-matching, and provided formal ADF bounds for the resulting approximation error in our rebuttal, along with a perturbative argument for the per-step Laplace error under Crank-Nicolson.
> > >
> > > More broadly, to our knowledge, analyzing complex inference pipelines component by component is the norm rather than the exception. INLA analyzes the Laplace approximation and the SPDE discretization separately; the extended Kalman filter literature analyzes linearization error and filter stability under separate assumptions. End-to-end theory combining all components of a multi-stage pipeline, where it exists at all, typically arrives in follow-up work.
> > >
> > > We would welcome specific guidance on which claim in our paper the reviewer considers unsupported by the combination of our theoretical results and empirical evidence. We have provided: (i) formal bounds on the novel approximation in our pipeline (moment-matching), (ii) a perturbative analysis of the Laplace error, (iii) empirical scaling matching the theoretical $O(N_s^{3/2})$ rate, (iv) a 200-instance calibration study, and (v) Gauss-Newton convergence on a nonlinear inverse problem. We believe this constitutes a thorough analysis by the standards of the field. If the sole remaining concern is the absence of a full end-to-end theorem --- which, as discussed above, remains an open problem in this space --- we would respectfully ask whether this should be grounds for rejection of a paper that makes a clear empirical and methodological advance.

---

### Decision · Program_Chairs · 2026-04-30

**Decision:**

Accept (regular)

**Comment:**

The paper presents a technically strong and timely contribution to probabilistic numerics by extending GP-based finite-volume methods to nonlinear conservation laws while substantially improving scalability. Across the reviews, there is clear agreement that the method is mathematically sound, original within its area, and empirically promising.

The rebuttal added a nonlinear inverse-problem experiment, stronger scalability evidence, and a substantially improved discussion of uncertainty calibration and model limitations. These additions addressed many of the initial concerns and led multiple reviewers to raise their scores.

The remaining reservations are mainly about framing rather than core technical validity. In particular, one reviewer still notes that the strongest claims around principled Bayesian uncertainty propagation and classical-FVM-style convergence are broader than what is fully established by the current end-to-end theory, even after rebuttal. I find this concern valid, and the final version should present these claims more cautiously. That said, the paper’s methodological advance and empirical evidence appear strong enough for acceptance.

Overall, this is a solid contribution that is likely to be useful to the probabilistic numerics community and to stimulate follow-up work on scalable uncertainty-aware PDE solvers.